# SMACv2: An Improved Benchmark for Cooperative Multi-Agent Reinforcement Learning

**Benjamin Ellis**[1][*] **Jonathan Cook**[1] **Skander Moalla**[1,5] **Mikayel Samvelyan**[2,3]
**Mingfei Sun**[4] **Anuj Mahajan**[1] **Jakob N. Foerster**[1] **Shimon Whiteson**[1]

[1]University of Oxford [2]University College London [3]Meta AI [4]University of Manchester
[5]EPFL

## Abstract

The availability of challenging benchmarks has played a key role in the recent progress of machine learning. In cooperative multi-agent reinforcement learning, the StarCraft Multi-Agent Challenge (SMAC) has become a popular testbed for the centralised training with decentralised execution paradigm. However, after years of sustained improvement on SMAC, algorithms now achieve near-perfect performance. In this work, we conduct new analysis demonstrating that SMAC lacks the stochasticity and partial observability to require complex *closed-loop* policies (i.e., those that condition on the observation). In particular, we show that an *open-loop* policy conditioned only on the timestep can achieve non-trivial win rates for many SMAC scenarios. To address this limitation, we introduce SMACv2, a new benchmark where scenarios are procedurally generated and require agents to generalise to previously unseen settings during evaluation.[2] We show that these changes ensure the benchmark requires the use of *closed-loop* policies. We also introduce the extended partial observability challenge (EPO), which augments SMACv2 to ensure meaningful partial observability. We evaluate state-of-the-art algorithms on SMACv2 and show that it presents significant challenges not present in the original benchmark. Our analysis illustrates that SMACv2 addresses the discovered deficiencies of SMAC and can help benchmark the next generation of MARL methods. Videos of training are available on our website.

## 1 Introduction

In many real-world cases, control policies for cooperative multi-agent reinforcement learning (MARL) can be learned in a setting where the algorithm has access to all agents' observations, e.g., in a simulator or laboratory, but must be deployed where such centralisation is not possible. Centralised training with decentralised execution (CTDE) [10, 25] is a paradigm for tackling these settings and has been the focus of much recent research [32, 44, 12, 26, 27]. In CTDE, the learning algorithm sees all observations during training, as well as possibly the Markov state, but must learn policies that can be executed without such privileged information.

As in other areas of machine learning, benchmarks have played an important role in driving progress in CTDE. Examples of such benchmarks include the Hanabi Learning Environment [2], Google football [20], PettingZoo [46], and Multi-Agent Mujoco [31]. Perhaps the most popular one is the StarCraft Multi-Agent Challenge (SMAC) [37], which focuses on decentralised micromanagement challenges in the game of StarCraft II. Rather than tackling the full game with centralised control [50], SMAC tasks a group of learning agents, each controlling a single army unit, to defeat the units of the

---

[*]Correspondence to benellis@robots.ox.ac.uk.

[2]Code is available at `https://github.com/oxwhirl/smacv2`

37th Conference on Neural Information Processing Systems (NeurIPS 2023) Track on Datasets and Benchmarks.

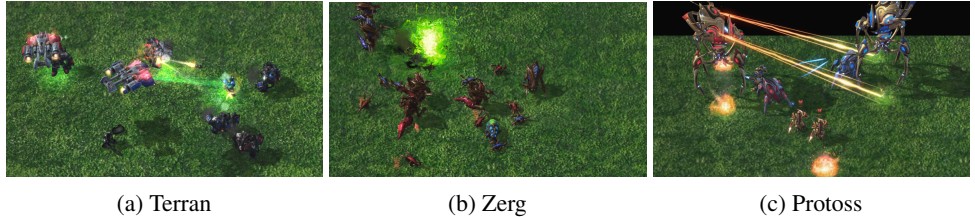

(a) Terran          (b) Zerg          (c) Protoss

Figure 1: Screenshots from SMACv2 showing agents battling the built-in AI.

enemy army controlled by the built-in heuristic AI. SMAC requires learning several complex joint action sequences such as focus fire,[3] and kiting enemy units,[4]. For these reasons, many prominent MARL papers [51, 26, 52, 33, 9, 55, 53] rely on SMAC to benchmark their performance.[5]

However, there is significant evidence that SMAC is outliving its usefulness [55, 13, 11]. Recent work reports near-perfect win rates on most scenarios [55, 13] and strong performance without using the centralised state via independent learning [55, 8]. This suggests that, due to ceiling effects, SMAC may no longer reward further algorithmic improvements and that new benchmarks are needed.

In this paper, we present new, more fundamental problems with SMAC and hence reason for new benchmarks. In particular, we show that an open-loop policy, which conditions only on the timestep and ignores all other observations, performs well on many SMAC scenarios. We also extend this analysis by examining the estimates of the joint $Q$-function learnt by QMIX. We demonstrate that, even with all features masked, regression to the joint $Q$-function is possible with less than 10% error in all but two scenarios from just timestep information.

Together these results suggest that SMAC is not stochastic enough to necessitate complex closed-loop (i.e. conditioned on the observation) control policies on many scenarios. Therefore, although SMAC scenarios may require difficult to discover action sequences such as focus fire, agents do not need to adapt to a diverse range of situations, but can largely repeat a fixed action sequence with little deviation. Additionally, *meaningful partial observability* (see Section 4) is minimal in SMAC due to large fields-of-view. For partial observability to be meaningful, one agent must observe information that is relevant to the current or future action selection of another agent, unknown to that other agent, and uninferrable from the other agent's observation. *Meaningful partial observability* is crucial to decentralisation and the NEXP-completeness of Dec-POMDPs [3].

To address these shortcomings, we propose *SMACv2*, a new benchmark that uses procedural content generation (PCG) [36] to address SMAC's lack of stochasticity. In SMACv2, for each episode we randomly generate team compositions and agent start positions. Consequently, it is no longer sufficient for agents to repeat a fixed action sequence, but they must learn to coordinate across a diverse range of scenarios. To remedy the lack of meaningful partial observability, we introduce the extended partial observability challenge (EPO), where we only allow the first agent to spot an enemy to see it for certain when it is in range. This requires agents to implicitly communicate enemy information to prioritise targets. Additionally, we update the sight and attack ranges to increase the diversity of the agents and complexity of scenarios.

Our implementation is extensible, allowing researchers to implement new distributions easily and hence further expand on the available SMACv2 scenarios. We also analyse SMACv2 and demonstrate that the added stochasticity prevents an open-loop policy from learning successfully, but instead requires policies to condition on ally and enemy features.

We evaluate several state-of-the-art algorithms on SMACv2 and show that they struggle with many scenarios, confirming that SMACv2 poses substantial new challenges. Finally, we perform an ablation on the new observation features to demonstrate how each contributes to the difficulty, highlighting the specific areas on which future MARL research should focus.

---

[3]In focused fire, allied units jointly attack and kill enemy units one after another.

[4]Kiting draws enemy units to give chase while maintaining distance so that little damage is incurred.

[5]SMAC has been cited over 600 times on Google Scholar.

## 2 Related Work

The MARL community has made significant use of games for benchmarking cooperative MARL algorithms. The Hanabi Learning Environment [2] tasks agents with learning to cooperate in the card-game Hanabi. Here, each agent observes the cards of its teammates but not its own, which must be implicitly communicated via gameplay. Hanabi is partially observable and stochastic, but only features teams of 2 to 5 players, which is fewer than all but the smallest SMACv2 scenarios. Kurach et al. [20] propose Google Football as a complex and stochastic benchmark, and also have a multi-agent setting. However, it assumes fully observable states, which simplifies coordination.

Peng et al. [31] propose a multi-agent adaptation of the MuJoCo environment featuring a complex continuous action space. Multi-Particle Environments (MPE) [25] feature simple communication-oriented challenges where particle agents can move and interact with each other using continuous actions. In contrast to both multi-agent MuJoCo and MPE, SMACv2 has a discrete action space, but challenges agents to handle a wide range of scenarios using procedural content generation. A plethora of work in MARL uses grid-worlds [25, 23, 54, 4, 48, 35, 24] where agents move and perform a small number of discrete actions on a 2D grid, but these tasks have much lower dimensional observations than SMACv2. OpenSpiel [22] and PettingZoo [46] provide collections of cooperative, competitive, and mixed sum games, such as grid-worlds and board games. However, the cooperative testbeds in either of these suites feature only simple environments with deterministic dynamics or a small number of agents. Neural MMO [41] provides a massively multi-agent game environment with open-ended tasks. However, it focuses on emergent behaviour within a large population of agents, rather than the fine-grained coordination of fully cooperative agents. Furthermore, none of these environments combines meaningful partial observability, complex dynamics, and high-dimensional observation spaces, whilst also featuring more than a few agents that need to coordinate to solve a common goal.

StarCraft has been frequently used as a testbed for RL algorithms. Most work focuses on the full game whereby a centralised controller serves as a puppeteer issuing commands for the two elements of the game: *macromanagement*, i.e., the high-level strategies for resource management and economy, and *micromanagement*, i.e., the fine-grained control of army units. TorchCraft [45] and TorchCraftAI [1] provide interfaces for training agents on *StarCraft: BroodWar*. The StarCraft II Learning Environment (SC2LE) [49] provides a Python interface for communicating with the game of *StarCraft II* and has been used to train AlphaStar [50], a grandmaster-level but fully centralised agent that is able to beat professional human players. SMAC and SMACv2 are built on top of SC2LE and concentrate only on decentralised unit micromanagement for the CTDE setting.

One limitation of SMAC is the constant starting positions and types of units, allowing methods to memorise action sequences for solving individual scenarios (as we show in Section 5.1), while also lacking the ability to generalise to new settings at test time, which is crucial for real-world applications of MARL [28]. To address these issues, SMACv2 relies on procedural content generation [PCG; 36, 18] whereby infinite game levels are generated algorithmically and differ across episodes. PCG environments have recently gained popularity in single-agent domains [7, 6, 17, 21, 38] for improving generalisation in RL [19] and we believe the next generation of MARL benchmarks should follow suit. Iqbal et al. [15] and Mahajan et al. [28] consider updated versions of SMAC by randomising the number and types of the units, respectively, to assess the generalisation in MARL. However, these works do not include the random start positions explored in SMACv2, analyse the properties of SMAC to motivate these changes, or address SMAC's lack of meaningful partial observability. They also do not change the agents' field-of-view and attack ranges or provide a convenient interface to generate new distributions over these features.

## 3 Background

### 3.1 Dec-POMDPs

A partially observable, cooperative multi-agent reinforcement learning task can be described by a *decentralised partially observable Markov decision process (Dec-POMDP)* [30]. This is a tuple $(n, S, A, T, \mathbb{O}, O, R, \gamma)$ where $n$ is the number of agents, $S$ is the set of states, $A$ is the set of individual actions, $T : S \times A^n \to \Delta(S)$ is the transition probability function, $\mathbb{O}$ is the set of joint observations, $O : S \times A^n \to \Delta(\mathbb{O})$ is the observation function, $R : S \times A^n \to \Delta(\mathbb{R})$ is the reward function, and $\gamma$ is the discount factor. We use $\Delta(X)$ to denote the set of probability distributions over a set $X$. At

each timestep $t$, each agent $i \in \{1, \ldots, n\}$ chooses an action $a \in A$. The global state then transitions from state $s \in S$ to $s' \in S$ and yields a reward $r \in \mathbb{R}$ after the joint action $\mathbf{a}$, according to the distribution $T(s, \mathbf{a})$ with probability $\mathbb{P}(s'|s, \mathbf{a})$ and $R(s, \mathbf{a})$ with probability $\mathbb{P}(r'|s, \mathbf{a})$, respectively. Each agent then receives an observation according to the observation function $O$ so that the joint observation $\mathbf{o} \in \mathbb{O}$ is sampled according to $O(s, \mathbf{a})$ with probability $\mathbb{P}(\mathbf{o}|s, \mathbf{a})$. This generates a trajectory $\tau_i \in (\mathbb{O} \times A^n)^*$. The goal is to learn $n$ policies $\pi_i : (\mathbb{O}_i \times A)^* \to \Delta(A)$ that maximise the expected cumulative reward $\mathbb{E}[\sum_i \gamma^i r_{t+i}]$.

### 3.2 StarCraft Multi-Agent Challenge (SMAC)

Rather than tackling the full game, the StarCraft Multi-Agent Challenge [37, SMAC] benchmark focuses on micromanagement challenges where each military unit is controlled by a single learning agent. Units have a limited field-of-view and no explicit communication mechanism when at test time. By featuring a set of challenging and diverse scenarios, SMAC has been used extensively by the MARL community for benchmarking algorithms. It consists of 14 micromanagement scenarios, which can be broadly divided into three different categories: *symmetric*, *asymmetric*, and *micro-trick*. The symmetric scenarios feature the same number and type of allied units as enemies. The asymmetric scenarios have one or more extra units for the enemy and are often more difficult than the symmetric scenarios. The micro-trick scenarios feature setups that require specific strategies to counter. For example, `3s_vs_5z` requires the three allied stalkers to kite the five zealots, and `corridor` requires six zealots to block off a narrow corridor to defeat twenty-four zerglings without being swarmed from all sides. Two state-of-the-art algorithms on SMAC are MAPPO [55] and QMIX [32]. We provide background to these methods in Section 2 of the appendix.

## 4 Partial Observability and Stochasticity in CTDE

If the initial state and transition dynamics of a Dec-POMDP are deterministic, then open-loop policies suffice, i.e., there exists an optimal joint policy that ignores all information but the timestep and agent ID. Such a policy amounts to playing back a fixed sequence of actions. **One way of determining whether SMAC has sufficient stochasticity therefore is to measure how strongly the learned policies condition on their observations**. For example, in a mostly deterministic environment, a policy may only be required to condition on a small subset of the observation space for a few timesteps, whereas a more stochastic environment might require conditioning on a large number of observation features. We use this concept in section 5.2 to evaluate SMAC's stochasticity.

Since open-loop policies do not rely on observations, the observation function and any partial observability it induces only become relevant when the environment is stochastic enough to require closed-loop policies. However, even in stochastic environments not all partial observability is meaningful. If the hidden information is not relevant to the task, or can be inferred from the observation, it is resolvable by a single agent alone, and if it is unknown and undiscoverable by any of the agents, then it cannot contribute to solving the task. **Without meaningful partial observability, a task cannot be considered truly decentralised because a policy can either ignore or infer any unknown information** *without the intervention of other agents*. Meaningful partial observability is essential to the difficulty (and NEXP-completeness) of Dec-POMDPs [3]. While SMAC is technically partially observable, there is limited hidden information because large fields-of-view render most relevant observations common among agents.

## 5 Limitations of SMAC

In this section, we analyse stochasticity in SMAC by examining the performance of open-loop policies and the predictability of Q-values given minimal observations. We show that SMAC is insufficiently stochastic to require complex closed-loop policies.

### 5.1 SMAC Stochasticity

We use the observation that if the initial state and transition functions are deterministic, there always exists an optimal deterministic policy that conditions solely on the timestep information to investigate stochasticity in SMAC. In particular, we use a range of SMAC scenarios to compare

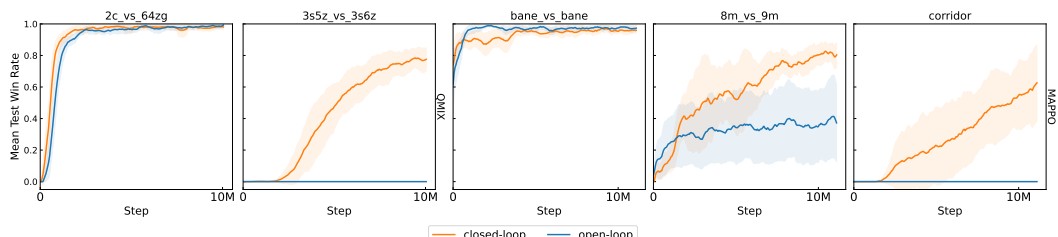

Figure 2: QMIX (left) and MAPPO (right) open-loop and closed-loop results.

policies conditioned only on the timestep (which we refer to as *open-loop*) to policies conditioned on the full observation history (i.e., all the information typically available to each agent in SMAC), which we refer to as *closed-loop*. The open-loop policies observe no ally or enemy information, and can only learn a distribution of actions at each timestep. If stochasticity in the underlying state is a serious challenge when training policies on SMAC scenarios, then policies without access to environment observations should fail. However, if the open-loop policies can succeed at the tasks, this demonstrates that SMAC is not representing the challenges of Dec-POMDPs well and so would benefit from added stochasticity.

We use MAPPO and QMIX as our base algorithms because of their widespread adoption and very strong performance and train open- and closed-loop versions of each. We train the *open-loop* policies on SMAC, but only allow the policies to observe the agent ID and timestep, whereas the closed-loop policies are given the usual SMAC observation as input with the timestep appended. The open-loop policies still have access to the central state as usual during training since this is only used in CTDE to aid in training and not during execution. The QMIX open-loop policy was trained only on the maps which the MAPPO open-loop policy was not able to completely solve. For QMIX, the hyperparameters used are the same as in [13] and for MAPPO the hyperparameters are listed in Appendix C.

Figure 2 shows the mean win rates and standard deviation of both policies on selected scenarios. The full results are available in Figure 8 in the appendix. Overall, the open-loop policies perform significantly worse than their closed-loop counterparts. However, this performance difference varies widely across scenarios and algorithms. For MAPPO, some scenarios, such as `bane_vs_bane`, `3s5z`, `1c3s5z` and `2s3z` achieve open-loop performance indistinguishable from that of closed-loop. In other scenarios, such as `8m_vs_9m` and `27m_vs_30m`, there are large performance differences, but the open-loop policy still achieves high win rates. There are also scenarios, such as `corridor` and `2c_vs_64zg`, where the MAPPO open-loop policy does not learn at all, whereas the closed-loop policies can learn successfully. QMIX also achieves good performance on `2c_vs_64zg`, `MMM2`, `27m_vs_30m` and `8m_vs_9m`, although for the latter maps the closed-loop policy strongly outperforms the open-loop QMIX. Altogether, there are only four SMAC maps where the open-loop approach cannot learn a good policy at all: `3s5z_vs_3s6z`, `corridor`, `6h_vs_8z` and `5m_vs_6m`. Overall, open-loop policies perform well on a range of scenarios in SMAC. The *easy* scenarios show no difference between the closed-loop and the open-loop policies, and some *hard* and *very hard* scenarios show either little difference or non-trivial win rates for the open-loop policy. These successes are striking given the restrictive nature of the open-loop policy. This suggests that stochasticity is not a significant challenge for a wide range of scenarios in SMAC.

These results highlight an important deficiency in SMAC. Stochasticity is either not evaluated or not part of the challenge for the vast majority of SMAC maps. Not testing stochasticity is a major flaw for a general MARL benchmark because without stochasticity the policy only needs to take optimal actions along a single trajectory. Additionally, without stochasticity, there can be no meaningful partial observability. Given widespread use of SMAC as a benchmark to evaluate algorithms for Dec-POMDPs, this suggests that a new benchmark is required to evaluate MARL algorithms.

## 5.2 SMAC Feature Inferrability & Relevance

In this section, we look at stochasticity from a different perspective. We mask (i.e. "zero-out") all features of the state and observations on which QMIX's joint Q-function conditions. If the Q-values can still be easily inferred via regression, then SMAC does not require learning complex closed-loop

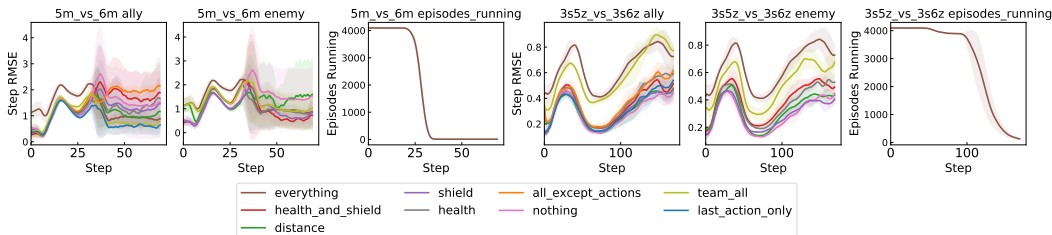

Figure 3: Comparison of the loss for different feature masks when regressing to a trained QMIX policy. The x-axis plots steps within an episode. 'Episodes Running' is the number of episodes that have not terminated by the given timestep. *everything* masks the entire observation and *nothing* masks no attributes. *team_all* masks all attributes of the team. *last_action_only* masks the last actions of all the allies in the state. The *all_except_actions* masks all the allied information *except* the last actions in the state. The mean is plotted for each mask and standard deviation across 3 seeds used as error bars. The low error rates for the everything mask imply that Q-values can be effectively inferred given only the timestep.

policies as Dec-POMDPs in general do, but is solvable by ignoring key aspects of StarCraft II gameplay. If the central $Q$-value could be inferred only from timestep information, this would also imply the task is not decentralised because all agents implicitly know the timestep and so could operate with the same central policy.

We also extend this experiment to masking subsets of the observation and state features. By only masking a subset of features, we measure the impact of different features on the learned policy. The more observation features the $Q$-function conditions on, the more confident we can be that SMAC requires understanding decentralised agent micromanagement rather than learning action sequences. We measure this conditioning by regressing to Q-values of a trained policy when masking different sets of features. The observation features hidden by each mask are in Table 1 in the appendix. Since we evaluate the regression performance of the joint Q-function, we always mask features in the state as well as the observation. More details of the regression procedure are in Appendix C.

Some interesting trends can be observed in Figure 3, which shows Q-value regression loss for each mask and scenario against steps in the RL episode. First, the root-mean-squared error of masking *everything* is low, reaching about $15\%$ of the mean $Q$-value at peak, and 5-10% for most of the episode. Mean root-mean-squared error as a proportion of mean $Q$-value, as well as other metrics, is given in the appendix in Table 2. For all scenarios except `5m_vs_6m` this value is below $0.12$. These are not much higher than the baseline *nothing* masks, suggesting the observation features do not really inform the Q-value. These results highlight the lack of stochasticity and meaningful partial observability in SMAC and the necessity of novel benchmarks to address these shortcomings.

## 6   SMACv2

As shown in the previous section, SMAC has some significant drawbacks. To address these, we propose three major changes: random team compositions, random start positions, and increasing diversity among unit types by using the true unit attack and sight ranges. These changes increase stochasticity and so address the deficiency discovered in the previous section.

Teams in SMACv2 are procedurally generated. Since StarCraft II does not allow units from different races to be on the same team, scenarios are divided depending on whether they feature Protoss, Terran or Zerg units. In scenarios with the same number of enemy and ally units, the unit types on each team are the same. Otherwise the teams are the same except for the extra enemy units, which are drawn identically to the other units. We use three unit types for each race, chosen to be the same unit types that were present in the original SMAC. We generate units in teams algorithmically, with each unit type having a fixed

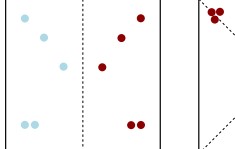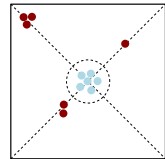

Figure 4: Examples of the two different types of start positions, *reflect* and *surround*. Allied units are shown in *blue* and enemy units in *dark red*.

probability. The probabilities of generating each unit type are given in Table 9 in the appendix. These probabilities are the same at test time and train time.

Random start positions in SMACv2 come in two different flavours. In *reflect* scenarios, the allied units are spawned uniformly at random on one side of the scenario. The enemy positions are the allied positions reflected in the vertical midpoint of the scenario. This is similar to how units spawned in the original SMAC benchmark but without clustering them together. In *surround* scenarios, allied units are spawned at the centre and surrounded by enemies stationed along the four diagonals. Figure 4 illustrates both flavours of start position.

The final change is to the sight range and attack range of the units. In the original SMAC, all unit types had a fixed sight and attack range, allowing short- and long-ranged units to choose to attack at the same distance. This did not affect the real attack range of units within the game – a unit ordered to attack an enemy while outside its striking range would approach the enemy before attacking. We change this to use the values from SC2, rather than the fixed values. However, we impose a minimum attack range of 2 because using the true attack ranges for melee units makes attacking too difficult.

Taken together, these changes mean that before an episode starts, agents no longer know what their unit type or initial position will be. Consequently, by diversifying the range of different scenarios the agents might encounter, they are required to understand the observation and state spaces more clearly, which should render learning a successful open-loop policy not possible. We also define a convenient interface for defining distributions over team unit types and start positions. By only implementing a small `Distribution` class one can easily change how these are generated.

Akin to the original version of SMAC [37], we split the scenarios into *symmetric* and *asymmetric* groups. In the symmetric scenarios, allies and enemies have the same number and type of units. In the asymmetric scenarios, the enemies have some extra units chosen from the same distribution as the allies. There are 5 unit, 10 unit and 20 unit symmetric scenarios, and 10 vs 11 and 20 vs 23 unit aymmetric scenarios for each of the three races.

## 6.1 Extended Partial Observability Challenge

We now introduce the Extended Partial Observability (EPO) challenge, where we make additional modifications to SMACv2 to ensure an extension of SMAC with meaningful partial observability. While SMAC is clearly partially observable in the sense that agents do not observe the global state, this partial observability is not particularly meaningful. SMAC certainly occludes information that would be relevant to each agent through the use of fixed sight ranges. However, the ranges of attack actions are always less than sight ranges and so ally teams can do without communication by moving within firing range and attacking any observed enemies. Finding enemies is not itself a significant challenge, as ally and enemy units spawn closely together. Therefore, much of the information that is relevant to the action selection of a given agent is within its observation.

To create a benchmark setting with meaningful partial observability, we introduce two further changes to SMACv2. First, enemy observations are stochastically masked for each agent. Once an enemy has been observed for the first time within an episode, the agent that was first to see it is guaranteed to observe it as normal. If two or more agents observe an enemy that had not yet been observed on the same timestep, there is a random tie break. As soon as this initial sighting is made, a random binary draw occurs for all other agents, which determines whether an agent will be able to observe that enemy for the remainder of the episode if it enters the agent's sight range. Agents for which the draw is unsuccessful have their observations masked such that they cannot observe any information about that enemy for the rest of the episode, even if it enters their sight range. The draw has a tunable probability $p$ of success. We suggest $p = 0$ in the challenge. Figure 5 illustrates this setup.

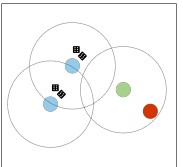 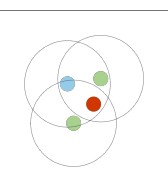

Figure 5: Example EPO enemy sighting. Allied units that do not observe the enemy are shown in *blue*, those that do are shown in *green* and the enemy unit in *dark red*. Initially, an ally spots an enemy. Later (right), when the enemy is within all allied sight ranges, only the first ally to observe the enemy and the ally for which the draw was successful can see it.

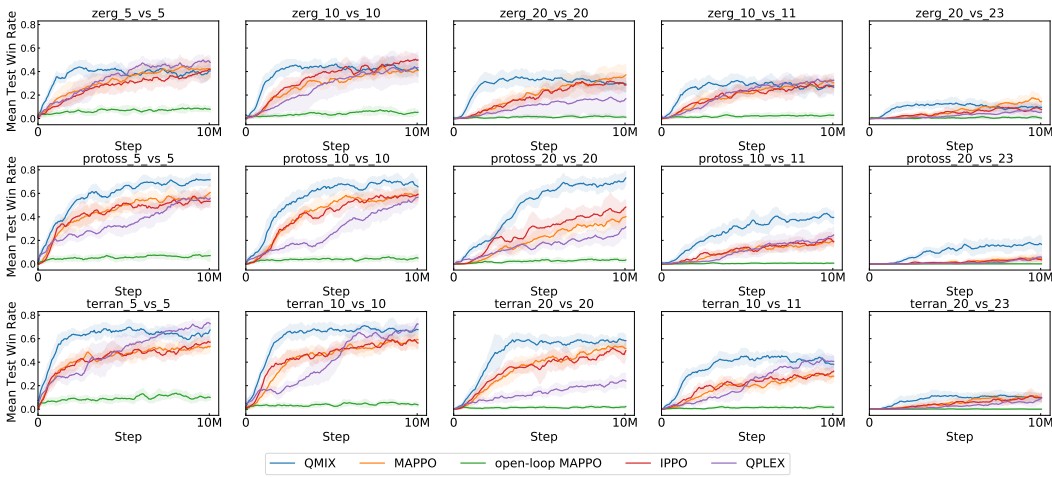

Figure 6: Comparison of the mean test win rate of QMIX, MAPPO, IPPO, QPLEX and an open-loop policy on SMACv2. Plots show the mean and standard deviation across 3 seeds.

In SMAC, the environment provides an available action mask that guarantees agents can only choose attack actions if the corresponding enemy is within shooting range. In this setting only, we remove this available actions mask. This makes deciding on a target more difficult because agents cannot rely on the available actions to infer which enemies they can attack. Hence they must either infer or communicate this information. If an invalid action is chosen, it becomes a no-op. We also found that without removing the available actions, there was little separation between the $p = 1$ (i.e. all allies can target all enemies) and $p = 0$ (i.e. only the first ally to see an enemy can target it) cases, as shown in Figure 13 in the appendix. We believe that this is because the available actions mask encourages heavy weight to be placed on attack actions, knowing that they will not be chosen unless available. In EPO, because of the increased difficulty, 6 allied agents battle against 5 enemy units. We found that this increased the gap between the $p = 1$ and $p = 0$ settings. We believe this is because in the more difficult `5_vs_5` setting, agents are too preoccupied with other more basic challenges for target prioritisation to be a large issue. Results for the `5_vs_5` setting are shown in Figure 14 in the appendix and for the `6_vs_5` setting in Figure 7. In EPO, there are 3 maps, one for each race, with 6 allies and 5 enemies per team. All other settings such as unit types and start positions are identical to SMACv2. We recommend a setting of $p = 0$.

# 7 SMACv2 Experiments

In this section, we present results from a number of experiments on SMACv2. We first train state-of-the-art SMAC baselines on SMACv2 to assess its difficulty. Currently, both value-based and policy optimisation approaches achieve strong performance on all SMAC scenarios [55, 13]. As baselines we use QMIX [32], a strong value-based method and MAPPO [55], a strong policy optimisation method. We also investigate the effect of the new SMACv2 environment features with ablations. These results are available in Figure 12 in the appendix.

## 7.1 SMACv2 Baseline Comparisons

First we compare the performance of MAPPO, QMIX, QPLEX[51] and IPPO[8] and an open-loop policy, based on MAPPO, on SMACv2. We run each algorithm for 10M training steps and 3 environment seeds. For MAPPO, we use the implementation from Sun et al. [43] and for QMIX the one from Hu et al. [13]. The results of the runs of the best hyperparameters are shown in Figure 6.

These results shown in Figure 6 reveal a few trends. First, QMIX generally performs better than MAPPO across most scenarios. QMIX strongly outperforms MAPPO in two Protoss scenarios (`protoss_20_vs_20` and `protoss_10_vs_11`), but otherwise final performance is similar, although QMIX is more sample efficient. However, QMIX is memory-intensive because of its large replay buffer and so requires more compute to run than MAPPO. Additionally, MAPPO appears to still be

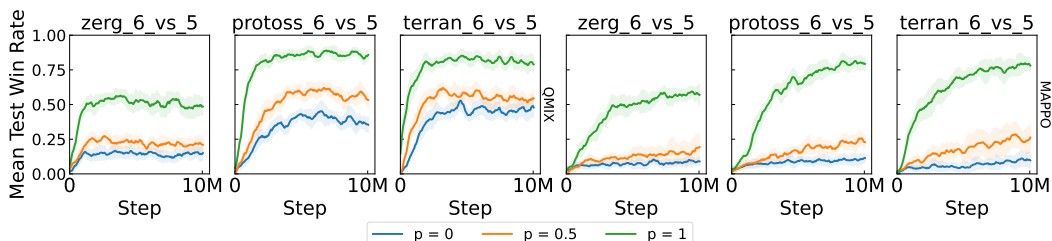

Figure 7: Comparison of mean test win rate when $p = 0, 0.5, 1$, with 6 agents against 5 enemy units, for QMIX (left) and MAPPO (right). Here, the available actions mask has been removed.

increasing its performance towards the end of training in several scenarios, such as the `10_vs_11` and `20_vs_20` maps. It is possible that MAPPO could attain better performance than QMIX if a larger compute budget were used. MAPPO and IPPO perform nearly identically across the maps, suggesting MAPPO gains little benefit from the centralised state. QPLEX performs worse than QMIX across a number of maps, particularly the `20_vs_20` maps.

Roughly even win rates are attained on the symmetric maps. By contrast, on the asymmetric maps, win rates are very low, particularly on the `20_vs_23` scenarios. Additionally, there is no real change in difficulty as the number of agents scales. However, the maps do not seem uniformly difficult, with all algorithms struggling with the Zerg scenarios more than other maps. Reassuringly the open-loop method cannot learn a good policy on any maps – even those where QMIX and MAPPO both achieve high win rates. This is strong evidence that SMAC's lack of stochasticity has been addressed.

## 7.2 EPO Baselines

We now compare QMIX and MAPPO on EPO. A comparison between the $p = 1$, $p = 0$ and $p = 0.5$ settings is shown in Figure 7. In this setting, $p = 0$ struggles to attain even modest performance, whereas $p = 1$ comes close to solving the task in a number of instances. This leaves a lot of room for the development of algorithms that learn to use implicit communication as a means of overcoming meaningfully restrictive partial observability and reduce the performance gap between the $p = 1$ and $p = 0$ settings. The performance in the $p = 0.5$ case is similar to the $p = 0$ case. This suggests that any communication protocol would have to be reasonably robust to have an impact on performance. This makes sense – an additional 50% chance to observe an enemy would still make it difficult to perform group target prioritisation. In general QMIX has a smaller gap between the $p = 0$ and $p = 1$ cases, suggesting it is perhaps better able to use the global state to resolve partial observability differences than MAPPO.

## 7.3 SMACv2 Feature Inferrability & Relevance

In this section we repeat the experiments from Section 5.2 on the 5 unit SMACv2 scenarios. We use three previously trained QMIX policies to generate 3 different datasets for training, and compute error bars in the regression across these three policies. The results are given in Table 3 and Figure 11 in the appendix.

When comparing results between SMAC and SMACv2, we focus on the difference between the *everything* (all features not visible) and *nothing* (no features hidden) masks. This is to account for some scenarios having much higher errors in the *nothing* mask than others. For example, `5m_vs_6m` has an error as a proportion of the mean Q-value in the *nothing* mask of $0.14$, compared to $0.018$ for `2c_vs_64zg`. The SMACv2 scenarios have higher errors than any of the SMAC scenarios when subtracting $\frac{\epsilon_{\text{rmse}}}{Q}$ for the *everything* and *nothing* masks. For example, this value is $0.12$ for `5_gen_protoss`, which has the lowest value for any tested SMACv2 map. This is significantly higher than the $0.07$ for `5m_vs_6m`, which has the highest value for any SMAC map. Tables 2 and 3 in the appendix list the full values for SMAC and SMACv2. These results suggest that SMACv2 is significantly more stochastic than SMAC. Second, examining stepwise results in Figure 11 in the appendix shows that SMACv2 maps have similar patterns of feature importance to SMAC maps. Enemy health is the most important individual feature and is important throughout an episode for all scenarios tested. Importantly, both the ally and enemy all masks reach significant thresholds of the

average $Q$-value, suggesting that it is important in SMACv2 for methods to attend to environment features.

## 8    Conclusion

Using a set of novel experimental evaluations, we showed that the popular SMAC benchmark for cooperative MARL suffers from a lack of stochasticity and meaningful partial observability. We then proposed SMACv2 and repeated these evaluations, demonstrating significant alleviation of this problem. We also demonstrated that baselines that achieve strong performance on SMAC struggle to achieve high win rates on the new scenarios in SMACv2. We performed ablations and discovered that the stochasticity of the unit types and random start positions combine to explain the tasks' difficulty. We also introduced and evaluated current methods on the extended partial observability (EPO) challenge and demonstrated that meaningful partial observability is a significant contributor to the challenge in this environment.

Gorsane et al. [11] found that it was most common for MARL papers accepted at top conferences to only evaluate on one benchmark, relying on the multiple scenarios within that benchmark to provide sufficient variety. However, in addition to the issues shown in that paper with cherry-picking of scenarios within a benchmark, this evaluation protocol is vulnerable to issues in environment dynamics, such as SMAC's lack of stochasticity, and community-wide overfitting. Given the difficulty in designing both benchmarks and sufficiently diverse scenarios within them, we recommend that evaluating on *multiple benchmarks* should become the norm where possible to help avoid issues.

We hope that SMACv2 can contribute to MARL research as a significantly challenging domain capturing practical challenges.

## Acknowledgements

Benjamin Ellis and Jonathan Cook are supported by the EPSRC centre for Doctoral Training in Autonomous and Intelligent Machines and Systems EP/S024050/1. The experiments were made possible by a generous equipment grant from NVIDIA.

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

# A  Links

- SMACv2: https://github.com/oxwhirl/smacv2. MIT License.
- MAPPO implementation: https://github.com/benellis3/mappo.
- QMIX implementation: https://github.com/benellis3/pymarl2. Apache-2 License.

# B  Further Background

## B.1  StarCraft

StarCraft II is a real-time strategy game featuring 3 different races, Protoss, Terran and Zerg, with different properties and associated strategies. The objective is to build an army powerful enough to destroy the enemy's base. When battling two armies, players must ensure army units are acting optimally. This is called *micromanagement*. An important micromanagement strategy is *focus firing*, which is ordering all allied units to jointly target the enemies one by one to ensure that damage taken is minimised.

Another important strategy is *kiting*, where units flee from the enemy and then pick them off one by one as they chase.

## B.2  QMIX

QMIX can be thought of as an extension of DQN[29] to the Dec-POMDP setting. The joint optimal action is found by forcing the joint $Q$ to adhere to the individual global max (IGM) principle[40], which states that the joint action can be found by maximising individual agents' $Q_i$ functions:

$$\arg\max_{\boldsymbol{a}} Q(s, \boldsymbol{\tau}, \boldsymbol{a}) = \begin{cases} \arg\max_a Q_1(\tau_1, a_1) \\ \arg\max_a Q_2(\tau_2, a_2) \\ \dots \\ \arg\max_a Q_n(\tau_n, a_n) \end{cases}$$

This central $Q$ is trained to regress to a target $r + \gamma \hat{Q}(s, \boldsymbol{\tau}, \boldsymbol{a})$ where $\hat{Q}$ is a target network that is updated slowly. The central $Q$ estimate is computed by a mixing network, whose weights are conditioned on the state, which takes as input the utility function $Q_i$ of the agents. The weights of the mixing network are restricted to be positive, which enforces the IGM principle[40] by ensuring the central $Q$ is monotonic in each $Q_i$.

## B.3  Independent and Multi-agent PPO

Proximal Policy Optimisation (PPO) is a method initially developed for single-agent reinforcement learning which aims to address performance collapse in policy gradient methods. It does this by heuristically bounding the ratio of action probabilities between the old and new policies. To this end, it optimises the below objective function.

$$\mathbb{E}_{s \sim d^\pi, a \sim \pi} \left[ \min\left( \frac{\tilde{\pi}(a|s)}{\pi(a|s)} A^\pi(s, a), \text{clip}\left( \frac{\tilde{\pi}(a|s)}{\pi(a|s)}, 1 - \epsilon, 1 + \epsilon \right) A^\pi(s, a) \right) \right]$$

where $\text{clip}(t, a, b)$ is a function that outputs $a$ if $t < a$, $b$ if $t > b$ and $t$ otherwise.

Extending this to the multi-agent setting can easily done in two ways. The first is to use independent learning, where each agent treats the others as part of the environment and learns a critic using its observation and action history. The second is to use a centralised critic conditioned on the central state. This is called multi-agent PPO. Both independent PPO (IPPO) and multi-agent PPO (MAPPO) have demonstrated strong performance on SMAC[55, 8]. Note that we do not apply the observation and state changes suggested by Yu et al. [55] anywhere in this paper. This is because these changes were not implemented as a wrapper on top of SMAC, but instead by modifying SMAC directly, leading to the environment code becoming unmanageably complicated.

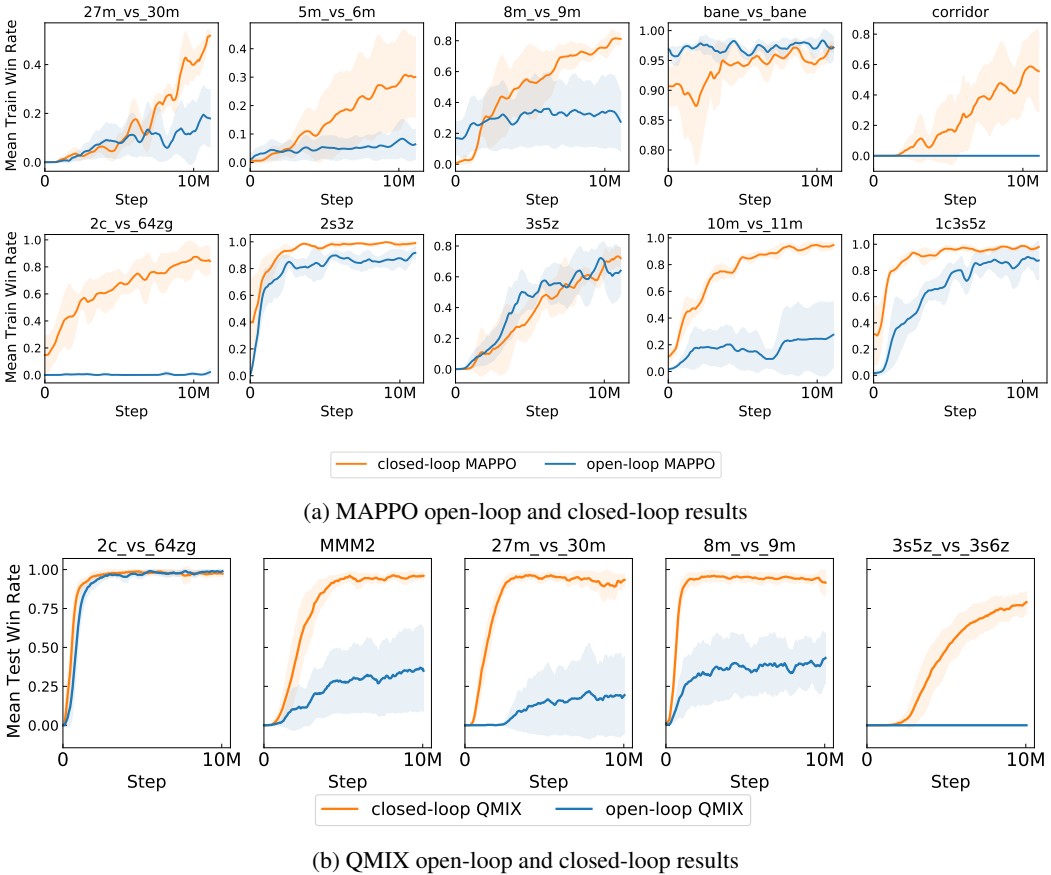

(a) MAPPO open-loop and closed-loop results

(b) QMIX open-loop and closed-loop results

Figure 8: Plot of selected SMAC scenarios treated as an open-loop planning problem by limiting the observation to the current timestep and agent ID. Plots show the mean win rate and standard deviation across 3 training seeds for MAPPO and QMIX.

## C  Experimental Details

In this section we describe extra details of the experiments run as part of the paper.

### C.1  Stochasticity

This section describes details of the experiments run in Section 5.1 in the main paper. Both the closed-loop and open-loop algorithms were based on the MAPPO implementation from Sun et al. [43]. The code used for these experiments can be found here. Both the open-loop and closed-loop algorithms use the same neural network architecture as in [43]. Both were also provided with access to the available actions mask of the environment and conditioned the critic on the state. We used the hyperparameters from Sun et al. [43], with the exception of the actor's learning rate, which we set to 0.0005 and decayed linearly throughout training. This is because learning rate decay has been found to be important for bounding PPO's policy updates [42]. Full hyperparameters can be found in Table 5. Full results are shown in Figure 8.

### C.2  SMAC Feature Inferrability & Relevance

This section describes details of the experiments run in Section 5.2 and Section 7.3 in the main paper. The code used for these experiments can be found here.

For each scenario, we had 3 QMIX policies trained using the implementation and hyperparameters from Hu et al. [13], each with a different network and SMAC initialization. Training results for the policies for SMAC are shown in Figure 9. Each policy constitutes a seed and is used to collect a

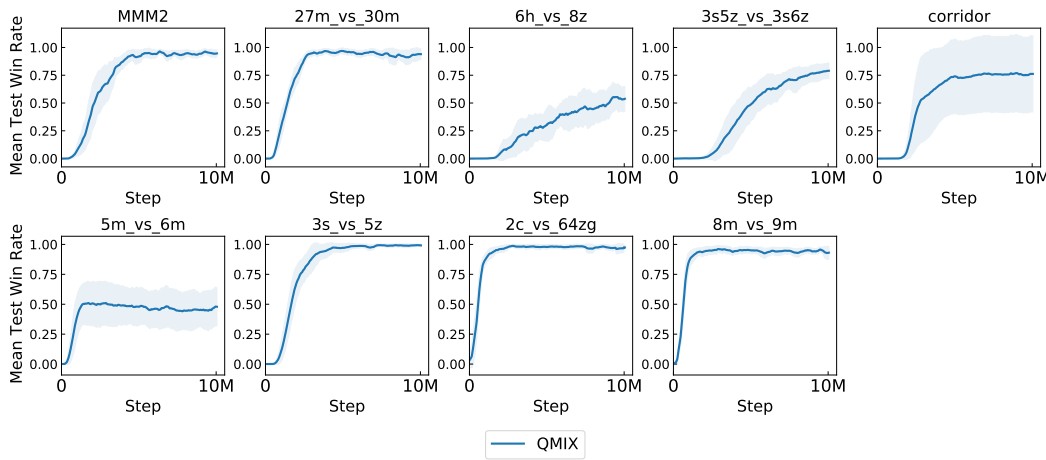

Figure 9: QMIX training results on SMAC

dataset of episodes for the feature-relevance experiment on the scenario. We call these policies *expert policies*. QMIX hyperparameters used are given in Table 4. A dataset consists of two folds: 8192 episodes used for training and 4096 episodes used for evaluation.

For a given mask, the experiment then consists of training a new QMIX network, termed *regression network*, whose input is trajectories with observations and states masked according to the mask, and whose task is to predict the $Q$-values output by the expert policy on the unmasked trajectories.

The *nothing* mask does not apply any effect on the observations and states in a trajectory. Otherwise, a mask, say 'health (ally)', masks the health feature of every ally in the observation of each agent (except its own health) and masks the health feature of all the units controlled by QMIX in the state. Table 1 shows the feature sets zeroed out for different masks.

The regression network has the same architecture as the expert network. We use the mean squared error (MSE) as a loss function and optimise the network via Adam with a batch size of 512 episodes and learning rate equal to 0.005. (The other Adam hyper-parameters are the default PyTorch values.) Training is performed with early stopping according to the validation fold with an evaluation every 5 epochs and a patience of 10 tries (i.e. training is stopped when the MSE on the validation fold does not decrease in 50 consecutive epochs and performance at the best epoch is retained). Models for all masks and scenarios hit early stopping and trained for about 200 epochs on average, except a few ones which trained to a limit of 500 epochs.

All hyper-parameters were tuned on a few different scenarios from both SMAC and SMACv2 to minimise validation MSE using datasets and expert policies not used for the reported results. The sizes of the training and validation folds are respectively 1.6 times and 0.8 times the size of the QMIX replay buffer (5000 episodes) and have been chosen to be large enough to minimize validation MSE while keeping experiments practical.

Figure 10 shows the full results of the feature quality experiments for SMAC and Figure 11 shows the full results for SMACv2. Table 3 shows summary results for the *everything* and *nothing* masks for SMACv2 and Table 2 shows the same data for SMAC.

Experiments for this section were conducted on 80-core CPU machines with NVIDIA GeForce RTX 2080 Ti or Tesla V100-SXM2-16GB GPUs. A single (scenario, seed, mask) combination took around 1 hour to train. In total, the experiments in this section took about 1500 GPU hours.

## C.3 SMACv2 Runs

Here we describe the hyperparameter and training procedure used in section 7.1. As in previous experiments, we used the implementation by Hu et al. [13] for QMIX and by Sun et al. [43] for MAPPO. These implementations have both achieved very strong results on SMAC. We therefore tuned the hyperparameters by taking deviations from these for a few key parameters. For MAPPO, we tuned the actor's learning rate and the clipping range. Additionally, we added linear learning rate

Table 1: Masked features for each setting of the feature quality experiment. ✓means a feature is masked and ✗means a feature is not masked.

| Mask | Ally | | | | | | Enemy | | | | |
|---|---|---|---|---|---|---|---|---|---|---|---|
| | Health | Shield | $x$ | $y$ | Distance | Actions | Health | Shield | $x$ | $y$ | Distance |
| *everything* | ✓ | ✓ | ✓ | ✓ | ✓ | ✓ | ✓ | ✓ | ✓ | ✓ | ✓ |
| *nothing* | ✗ | ✗ | ✗ | ✗ | ✗ | ✗ | ✗ | ✗ | ✗ | ✗ | ✗ |
| *health (ally)* | ✓ | ✗ | ✗ | ✗ | ✗ | ✗ | ✗ | ✗ | ✗ | ✗ | ✗ |
| *shield (ally)* | ✗ | ✓ | ✗ | ✗ | ✗ | ✗ | ✗ | ✗ | ✗ | ✗ | ✗ |
| *distance (ally)* | ✗ | ✗ | ✓ | ✓ | ✓ | ✗ | ✗ | ✗ | ✗ | ✗ | ✗ |
| *health and shield (ally)* | ✓ | ✓ | ✗ | ✗ | ✗ | ✗ | ✗ | ✗ | ✗ | ✗ | ✗ |
| *actions only* | ✗ | ✗ | ✗ | ✗ | ✗ | ✓ | ✗ | ✗ | ✗ | ✗ | ✗ |
| *all except actions* | ✓ | ✓ | ✓ | ✓ | ✓ | ✗ | ✗ | ✗ | ✗ | ✗ | ✗ |
| *ally all* | ✓ | ✓ | ✓ | ✓ | ✓ | ✓ | ✗ | ✗ | ✗ | ✗ | ✗ |
| *health (enemy)* | ✗ | ✗ | ✗ | ✗ | ✗ | ✗ | ✓ | ✗ | ✗ | ✗ | ✗ |
| *shield (enemy)* | ✗ | ✗ | ✗ | ✗ | ✗ | ✗ | ✗ | ✓ | ✗ | ✗ | ✗ |
| *distance (enemy)* | ✗ | ✗ | ✗ | ✗ | ✗ | ✗ | ✗ | ✗ | ✓ | ✓ | ✓ |
| *enemy all* | ✗ | ✗ | ✗ | ✗ | ✗ | ✗ | ✓ | ✓ | ✓ | ✓ | ✓ |

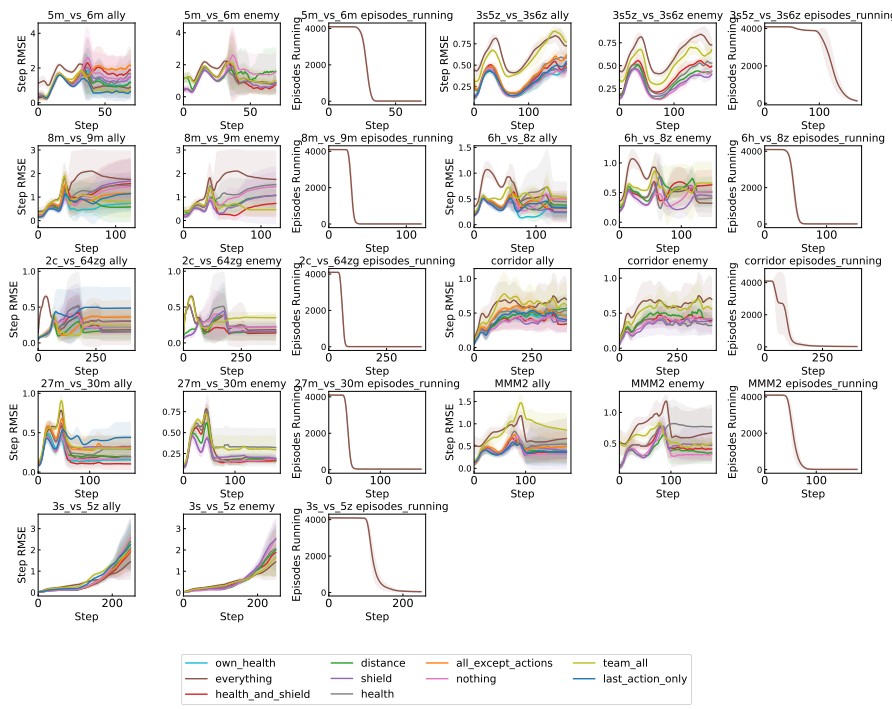

Figure 10: Results of the feature quality experiments for all tested SMAC scenarios not in the main paper.

Table 2: Mean Q values in the different feature quality experiments on SMAC scenarios

| Map | Mask | $\bar{Q}$ | $\epsilon_{\text{rmse}}$ | $\frac{\epsilon_{\text{rmse}}}{\bar{Q}}$ | $\epsilon_{\text{abs}}$ | $\frac{\epsilon_{\text{rmse}}^{\text{mask}} - \epsilon_{\text{rmse}}^{\text{nothing}}}{\bar{Q}}$ |
|---|---|---|---|---|---|---|
| corridor | everything | $7.0 \pm 0.08$ | $0.55 \pm 0.08$ | $0.078 \pm 0.002$ | $0.39 \pm 0.05$ | $0.042 \pm 0.001$ |
| | ally_all_except_actions | | $0.34 \pm 0.09$ | $0.047 \pm 0.005$ | $0.21 \pm 0.04$ | $0.01 \pm 0.008$ |
| | ally_health_and_shield | | $0.29 \pm 0.07$ | $0.041 \pm 0.003$ | $0.19 \pm 0.03$ | $0.005 \pm 0.006$ |
| | ally_health | | $0.29 \pm 0.05$ | $0.0413 \pm 0.0007$ | $0.19 \pm 0.02$ | $0.005 \pm 0.004$ |
| | ally_distance | | $0.27 \pm 0.03$ | $0.038 \pm 0.002$ | $0.174 \pm 0.009$ | $0.001 \pm 0.001$ |
| | ally_shield | | $0.26 \pm 0.02$ | $0.037 \pm 0.003$ | $0.166 \pm 0.005$ | $0.0004 \pm 0.0003$ |
| | ally_all | | $0.4 \pm 0.2$ | $0.06 \pm 0.01$ | $0.28 \pm 0.09$ | $0.02 \pm 0.02$ |
| | enemy_health_and_shield | | $0.29 \pm 0.05$ | $0.0414 \pm 0.0005$ | $0.2 \pm 0.03$ | $0.005 \pm 0.004$ |
| | enemy_health | | $0.3 \pm 0.04$ | $0.043 \pm 0.003$ | $0.21 \pm 0.02$ | $0.006 \pm 0.001$ |
| | enemy_distance | | $0.32 \pm 0.04$ | $0.046 \pm 0.002$ | $0.203 \pm 0.009$ | $0.009 \pm 0.001$ |
| | enemy_shield | | $0.25 \pm 0.03$ | $0.036 \pm 0.003$ | $0.161 \pm 0.005$ | $-0.0001 \pm 0.0008$ |
| | enemy_all | | $0.42 \pm 0.09$ | $0.059 \pm 0.003$ | $0.29 \pm 0.06$ | $0.022 \pm 0.006$ |
| | nothing | | $0.25 \pm 0.02$ | $0.037 \pm 0.003$ | $0.164 \pm 0.006$ | |
| 3s5z_vs_3s6z | everything | $6.97 \pm 0.01$ | $0.61 \pm 0.01$ | $0.087 \pm 0.001$ | $0.418 \pm 0.006$ | $0.046 \pm 0.003$ |
| | ally_all_except_actions | | $0.34 \pm 0.01$ | $0.049 \pm 0.002$ | $0.193 \pm 0.01$ | $0.0083 \pm 0.0009$ |
| | ally_health_and_shield | | $0.326 \pm 0.01$ | $0.047 \pm 0.002$ | $0.182 \pm 0.009$ | $0.006 \pm 0.001$ |
| | ally_health | | $0.333 \pm 0.006$ | $0.0479 \pm 0.0008$ | $0.188 \pm 0.001$ | $0.007 \pm 0.001$ |
| | ally_distance | | $0.29 \pm 0.02$ | $0.041 \pm 0.003$ | $0.16 \pm 0.01$ | $-0.0 \pm 0.003$ |
| | ally_shield | | $0.299 \pm 0.007$ | $0.043 \pm 0.001$ | $0.167 \pm 0.004$ | $0.0018 \pm 0.0004$ |
| | ally_all | | $0.53 \pm 0.02$ | $0.076 \pm 0.003$ | $0.35 \pm 0.02$ | $0.035 \pm 0.004$ |
| | enemy_health_and_shield | | $0.38 \pm 0.02$ | $0.055 \pm 0.003$ | $0.25 \pm 0.02$ | $0.014 \pm 0.002$ |
| | enemy_health | | $0.36 \pm 0.01$ | $0.051 \pm 0.002$ | $0.22 \pm 0.01$ | $0.01 \pm 0.001$ |
| | enemy_distance | | $0.32 \pm 0.01$ | $0.046 \pm 0.002$ | $0.18 \pm 0.008$ | $0.005 \pm 0.001$ |
| | enemy_shield | | $0.302 \pm 0.004$ | $0.0434 \pm 0.0007$ | $0.188 \pm 0.006$ | $0.0023 \pm 0.0008$ |
| | enemy_all | | $0.5 \pm 0.04$ | $0.071 \pm 0.007$ | $0.33 \pm 0.04$ | $0.03 \pm 0.006$ |
| | nothing | | $0.286 \pm 0.008$ | $0.041 \pm 0.001$ | $0.161 \pm 0.007$ | |
| 5m_vs_6m | everything | $8.1 \pm 0.08$ | $1.71 \pm 0.08$ | $0.21 \pm 0.02$ | $1.24 \pm 0.09$ | $0.073 \pm 0.007$ |
| | ally_all_except_actions | | $1.17 \pm 0.04$ | $0.15 \pm 0.02$ | $0.76 \pm 0.06$ | $0.007 \pm 0.004$ |
| | ally_health_and_shield | | $1.13 \pm 0.05$ | $0.14 \pm 0.02$ | $0.71 \pm 0.05$ | $0.002 \pm 0.002$ |
| | ally_health | | $1.13 \pm 0.05$ | $0.14 \pm 0.02$ | $0.72 \pm 0.05$ | $0.001 \pm 0.006$ |
| | ally_distance | | $1.15 \pm 0.05$ | $0.15 \pm 0.02$ | $0.73 \pm 0.05$ | $0.004 \pm 0.004$ |
| | ally_shield | | $1.12 \pm 0.07$ | $0.14 \pm 0.02$ | $0.71 \pm 0.05$ | $0.001 \pm 0.002$ |
| | ally_all | | $1.24 \pm 0.06$ | $0.16 \pm 0.02$ | $0.83 \pm 0.09$ | $0.014 \pm 0.007$ |
| | enemy_health_and_shield | | $1.28 \pm 0.03$ | $0.16 \pm 0.02$ | $0.81 \pm 0.03$ | $0.021 \pm 0.001$ |
| | enemy_health | | $1.27 \pm 0.05$ | $0.16 \pm 0.02$ | $0.82 \pm 0.06$ | $0.019 \pm 0.004$ |
| | enemy_distance | | $1.22 \pm 0.02$ | $0.15 \pm 0.02$ | $0.82 \pm 0.02$ | $0.013 \pm 0.007$ |
| | enemy_shield | | $1.11 \pm 0.06$ | $0.14 \pm 0.02$ | $0.7 \pm 0.06$ | $-0.0 \pm 0.003$ |
| | enemy_all | | $1.44 \pm 0.06$ | $0.18 \pm 0.02$ | $0.99 \pm 0.07$ | $0.04 \pm 0.009$ |
| | nothing | | $1.11 \pm 0.05$ | $0.14 \pm 0.02$ | $0.71 \pm 0.04$ | |
| 8m_vs_9m | everything | $12.3 \pm 0.07$ | $0.75 \pm 0.07$ | $0.061 \pm 0.006$ | $0.46 \pm 0.06$ | $0.025 \pm 0.004$ |
| | ally_all_except_actions | | $0.6 \pm 0.04$ | $0.049 \pm 0.004$ | $0.32 \pm 0.04$ | $0.013 \pm 0.003$ |
| | ally_health_and_shield | | $0.53 \pm 0.05$ | $0.043 \pm 0.005$ | $0.27 \pm 0.04$ | $0.007 \pm 0.002$ |
| | ally_health | | $0.56 \pm 0.03$ | $0.045 \pm 0.002$ | $0.29 \pm 0.02$ | $0.01 \pm 0.004$ |
| | ally_distance | | $0.49 \pm 0.06$ | $0.04 \pm 0.005$ | $0.27 \pm 0.05$ | $0.0039 \pm 0.0006$ |
| | ally_shield | | $0.47 \pm 0.02$ | $0.038 \pm 0.002$ | $0.27 \pm 0.02$ | $0.002 \pm 0.003$ |
| | ally_all | | $0.73 \pm 0.07$ | $0.06 \pm 0.007$ | $0.43 \pm 0.06$ | $0.024 \pm 0.004$ |
| | enemy_health_and_shield | | $0.55 \pm 0.04$ | $0.045 \pm 0.004$ | $0.31 \pm 0.03$ | $0.009 \pm 0.002$ |
| | enemy_health | | $0.55 \pm 0.05$ | $0.045 \pm 0.004$ | $0.32 \pm 0.03$ | $0.009 \pm 0.001$ |
| | enemy_distance | | $0.46 \pm 0.07$ | $0.038 \pm 0.006$ | $0.27 \pm 0.04$ | $0.002 \pm 0.002$ |
| | enemy_shield | | $0.42 \pm 0.05$ | $0.034 \pm 0.005$ | $0.23 \pm 0.04$ | $-0.001 \pm 0.002$ |
| | enemy_all | | $0.64 \pm 0.06$ | $0.052 \pm 0.006$ | $0.38 \pm 0.05$ | $0.016 \pm 0.003$ |
| | nothing | | $0.44 \pm 0.07$ | $0.036 \pm 0.006$ | $0.24 \pm 0.04$ | |
| 27m_vs_30m | everything | $9.6 \pm 0.04$ | $0.49 \pm 0.04$ | $0.051 \pm 0.006$ | $0.33 \pm 0.03$ | $0.016 \pm 0.002$ |
| | ally_all_except_actions | | $0.47 \pm 0.05$ | $0.049 \pm 0.007$ | $0.28 \pm 0.04$ | $0.014 \pm 0.003$ |
| | ally_health_and_shield | | $0.44 \pm 0.07$ | $0.046 \pm 0.009$ | $0.26 \pm 0.05$ | $0.01 \pm 0.003$ |
| | ally_health | | $0.45 \pm 0.06$ | $0.047 \pm 0.007$ | $0.26 \pm 0.04$ | $0.011 \pm 0.002$ |
| | ally_distance | | $0.35 \pm 0.06$ | $0.037 \pm 0.008$ | $0.21 \pm 0.04$ | $0.001 \pm 0.001$ |
| | ally_shield | | $0.35 \pm 0.02$ | $0.036 \pm 0.004$ | $0.2 \pm 0.02$ | $0.001 \pm 0.003$ |
| | ally_all | | $0.5 \pm 0.06$ | $0.052 \pm 0.008$ | $0.32 \pm 0.05$ | $0.016 \pm 0.003$ |
| | enemy_health_and_shield | | $0.46 \pm 0.04$ | $0.048 \pm 0.006$ | $0.31 \pm 0.03$ | $0.012 \pm 0.004$ |
| | enemy_health | | $0.5 \pm 0.05$ | $0.052 \pm 0.007$ | $0.34 \pm 0.04$ | $0.0166 \pm 0.0003$ |
| | enemy_distance | | $0.41 \pm 0.03$ | $0.042 \pm 0.005$ | $0.25 \pm 0.03$ | $0.007 \pm 0.003$ |
| | enemy_shield | | $0.35 \pm 0.06$ | $0.036 \pm 0.007$ | $0.21 \pm 0.03$ | $0.0009 \pm 0.0008$ |
| | enemy_all | | $0.53 \pm 0.02$ | $0.055 \pm 0.004$ | $0.37 \pm 0.02$ | $0.02 \pm 0.002$ |
| | nothing | | $0.34 \pm 0.05$ | $0.036 \pm 0.006$ | $0.21 \pm 0.04$ | |
| 2c_vs_64zg | everything | $7.72 \pm 0.02$ | $0.56 \pm 0.02$ | $0.072 \pm 0.002$ | $0.42 \pm 0.01$ | $0.054 \pm 0.003$ |
| | ally_all_except_actions | | $0.15 \pm 0.04$ | $0.02 \pm 0.005$ | $0.08 \pm 0.02$ | $0.0018 \pm 0.001$ |
| | ally_health_and_shield | | $0.15 \pm 0.04$ | $0.02 \pm 0.005$ | $0.08 \pm 0.02$ | $0.002 \pm 0.002$ |
| | ally_health | | $0.15 \pm 0.03$ | $0.02 \pm 0.004$ | $0.08 \pm 0.02$ | $0.0018 \pm 0.0005$ |
| | ally_distance | | $0.13 \pm 0.03$ | $0.017 \pm 0.003$ | $0.07 \pm 0.01$ | $-0.0007 \pm 0.0004$ |
| | ally_shield | | $0.14 \pm 0.02$ | $0.018 \pm 0.002$ | $0.07 \pm 0.01$ | $0.0 \pm 0.001$ |
| | ally_all | | $0.15 \pm 0.04$ | $0.019 \pm 0.005$ | $0.08 \pm 0.02$ | $0.002 \pm 0.001$ |
| | enemy_health_and_shield | | $0.43 \pm 0.006$ | $0.056 \pm 0.002$ | $0.317 \pm 0.004$ | $0.038 \pm 0.003$ |
| | enemy_health | | $0.426 \pm 0.006$ | $0.0553 \pm 0.001$ | $0.314 \pm 0.006$ | $0.037 \pm 0.003$ |
| | enemy_distance | | $0.18 \pm 0.02$ | $0.024 \pm 0.002$ | $0.125 \pm 0.009$ | $0.006 \pm 0.001$ |
| | enemy_shield | | $0.14 \pm 0.03$ | $0.018 \pm 0.003$ | $0.07 \pm 0.01$ | $0.0 \pm 0.0008$ |
| | enemy_all | | $0.54 \pm 0.02$ | $0.07 \pm 0.002$ | $0.41 \pm 0.01$ | $0.052 \pm 0.003$ |
| | nothing | | $0.14 \pm 0.03$ | $0.018 \pm 0.004$ | $0.07 \pm 0.02$ | |
| 6h_vs_8z | everything | $7.5 \pm 0.09$ | $0.87 \pm 0.09$ | $0.12 \pm 0.02$ | $0.65 \pm 0.09$ | $0.07 \pm 0.01$ |
| | ally_all_except_actions | | $0.43 \pm 0.02$ | $0.059 \pm 0.008$ | $0.31 \pm 0.03$ | $0.007 \pm 0.002$ |
| | ally_health_and_shield | | $0.41 \pm 0.03$ | $0.056 \pm 0.009$ | $0.29 \pm 0.03$ | $0.0034 \pm 0.0007$ |
| | ally_health | | $0.41 \pm 0.03$ | $0.055 \pm 0.008$ | $0.28 \pm 0.03$ | $0.003 \pm 0.002$ |
| | ally_distance | | $0.41 \pm 0.03$ | $0.056 \pm 0.009$ | $0.29 \pm 0.03$ | $0.004 \pm 0.001$ |
| | ally_shield | | $0.38 \pm 0.03$ | $0.051 \pm 0.008$ | $0.26 \pm 0.03$ | $-0.001 \pm 0.001$ |
| | ally_all | | $0.49 \pm 0.01$ | $0.067 \pm 0.007$ | $0.36 \pm 0.02$ | $0.014 \pm 0.002$ |
| | enemy_health_and_shield | | $0.45 \pm 0.02$ | $0.06 \pm 0.008$ | $0.32 \pm 0.02$ | $0.008 \pm 0.001$ |
| | enemy_health | | $0.46 \pm 0.03$ | $0.063 \pm 0.01$ | $0.33 \pm 0.03$ | $0.011 \pm 0.001$ |
| | enemy_distance | | $0.47 \pm 0.03$ | $0.064 \pm 0.009$ | $0.33 \pm 0.03$ | $0.012 \pm 0.0004$ |
| | enemy_shield | | $0.39 \pm 0.02$ | $0.053 \pm 0.008$ | $0.27 \pm 0.03$ | $0.001 \pm 0.002$ |
| | enemy_all | | $0.55 \pm 0.02$ | $0.075 \pm 0.008$ | $0.4 \pm 0.02$ | $0.0224 \pm 0.001$ |
| | nothing | | $0.38 \pm 0.03$ | $0.052 \pm 0.009$ | $0.27 \pm 0.03$ | |
| 3s_vs_5z | everything | $8.1 \pm 0.04$ | $0.3 \pm 0.04$ | $0.037 \pm 0.007$ | $0.2 \pm 0.03$ | $0.012 \pm 0.003$ |
| | ally_all_except_actions | | $0.19 \pm 0.02$ | $0.024 \pm 0.003$ | $0.1 \pm 0.02$ | $-0.0014 \pm 0.0008$ |
| | ally_health_and_shield | | $0.17 \pm 0.02$ | $0.021 \pm 0.002$ | $0.081 \pm 0.01$ | $-0.004 \pm 0.001$ |
| | ally_health | | $0.17 \pm 0.03$ | $0.021 \pm 0.004$ | $0.07 \pm 0.01$ | $-0.004 \pm 0.002$ |
| | ally_distance | | $0.2 \pm 0.02$ | $0.024 \pm 0.002$ | $0.09 \pm 0.02$ | $-0.0 \pm 0.004$ |
| | ally_shield | | $0.18 \pm 0.02$ | $0.022 \pm 0.002$ | $0.083 \pm 0.007$ | $-0.003 \pm 0.003$ |
| | ally_all | | $0.3 \pm 0.08$ | $0.04 \pm 0.01$ | $0.17 \pm 0.05$ | $0.012 \pm 0.007$ |
| | enemy_health_and_shield | | $0.26 \pm 0.03$ | $0.032 \pm 0.004$ | $0.17 \pm 0.03$ | $0.007 \pm 0.001$ |
| | enemy_health | | $0.24 \pm 0.03$ | $0.03 \pm 0.005$ | $0.15 \pm 0.03$ | $0.005 \pm 0.002$ |
| | enemy_distance | | $0.19 \pm 0.03$ | $0.024 \pm 0.003$ | $0.09 \pm 0.02$ | $-0.0015 \pm 0.0006$ |
| | enemy_shield | | $0.24 \pm 0.02$ | $0.03 \pm 0.004$ | $0.13 \pm 0.01$ | $0.0049 \pm 0.0002$ |
| | enemy_all | | $0.26 \pm 0.03$ | $0.033 \pm 0.004$ | $0.17 \pm 0.02$ | $0.008 \pm 0.001$ |
| | nothing | | $0.2 \pm 0.02$ | $0.025 \pm 0.003$ | $0.08 \pm 0.01$ | |
| MMM2 | everything | $9.4 \pm 0.1$ | $0.7 \pm 0.1$ | $0.08 \pm 0.02$ | $0.47 \pm 0.09$ | $0.039 \pm 0.008$ |
| | ally_all_except_actions | | $0.41 \pm 0.07$ | $0.04 \pm 0.01$ | $0.23 \pm 0.05$ | $0.007 \pm 0.003$ |
| | ally_health_and_shield | | $0.38 \pm 0.03$ | $0.041 \pm 0.005$ | $0.22 \pm 0.03$ | $0.003 \pm 0.002$ |
| | ally_health | | $0.39 \pm 0.04$ | $0.042 \pm 0.006$ | $0.21 \pm 0.04$ | $0.004 \pm 0.002$ |
| | ally_distance | | $0.34 \pm 0.05$ | $0.036 \pm 0.007$ | $0.19 \pm 0.04$ | $-0.0015 \pm 0.0007$ |
| | ally_shield | | $0.35 \pm 0.06$ | $0.038 \pm 0.008$ | $0.2 \pm 0.04$ | $-0.0001 \pm 0.0009$ |
| | ally_all | | $0.6 \pm 0.1$ | $0.07 \pm 0.02$ | $0.36 \pm 0.09$ | $0.028 \pm 0.008$ |
| | enemy_health_and_shield | | $0.43 \pm 0.06$ | $0.046 \pm 0.008$ | $0.29 \pm 0.04$ | $0.008 \pm 0.001$ |
| | enemy_health | | $0.44 \pm 0.05$ | $0.047 \pm 0.007$ | $0.29 \pm 0.04$ | $0.0094 \pm 0.0005$ |
| | enemy_distance | | $0.33 \pm 0.05$ | $0.036 \pm 0.006$ | $0.19 \pm 0.04$ | $-0.0022 \pm 0.0009$ |
| | enemy_shield | | $0.33 \pm 0.04$ | $0.035 \pm 0.006$ | $0.18 \pm 0.04$ | $-0.002 \pm 0.002$ |
| | enemy_all | | $0.59 \pm 0.08$ | $0.06 \pm 0.01$ | $0.39 \pm 0.05$ | $0.026 \pm 0.004$ |
| | nothing | | $0.35 \pm 0.05$ | $0.038 \pm 0.007$ | $0.2 \pm 0.04$ | |

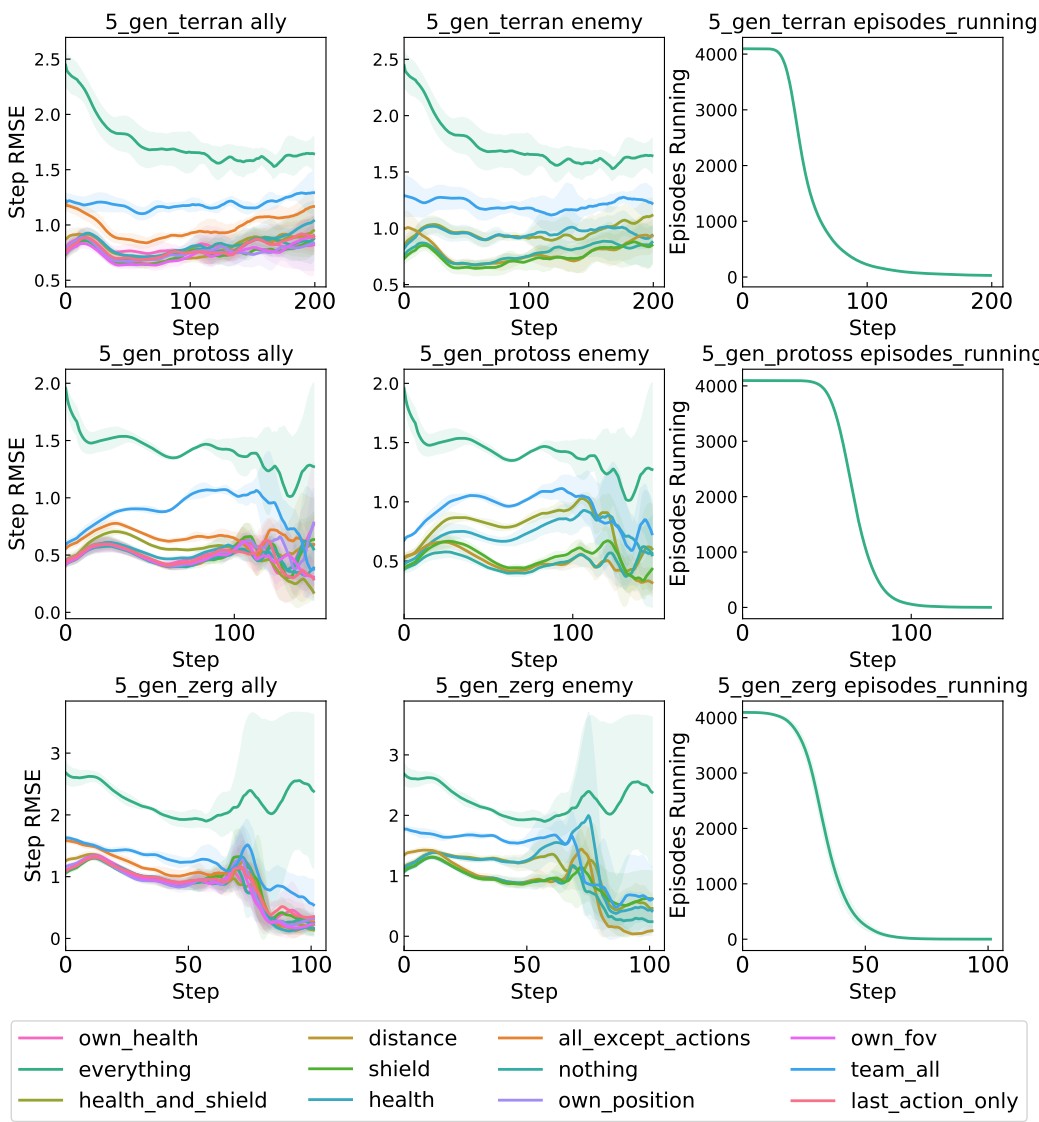

Figure 11: Feature Quality Experiments for the 5 and 10 unit SMACv2 scenarios.

decay to the MAPPO implementation because this has been shown to be important to bound the policy update[42]. For QMIX, we tuned the $\epsilon$-annealing time and $\lambda$ in the eligibility trace $Q(\lambda)$. All other hyperparameters, including neural network architecture, were unchanged from the implementations mentioned previously.

The MAPPO code for these experiments can be found here, and the QMIX code here. The QMIX code is distributed under the Apache license, and the MAPPO code under the MIT license. The only differences in these branches and the implementations used for previous experiments on SMAC are important to changing the environment from SMAC to SMACv2.

The open-loop policy is identical to MAPPO, except the policy receives as input only the timestamp and agent ID, as in Section 5.1.

We tuned hyperparameters by running each set of hyperparameters on all scenarios and then choosing the best results. The grids of hyperparameters for QMIX and MAPPO can be found in Table 7 and 8. For IPPO, we used mostly the same hyperparameters as MAPPO. For QPLEX we used the hyperparameters from [51]. All algorithms were trained for 10M environment steps with evaluations

Table 3: SMACv2 feature quality experiment results

| Map | Mask | $\bar{Q}$ | $\epsilon_{\text{rmse}}$ | $\frac{\epsilon_{\text{rmse}}}{\bar{Q}}$ | $\epsilon_{\text{abs}}$ | $\frac{\epsilon_{\text{rmse}}^{\text{mask}} - \epsilon_{\text{rmse}}^{\text{nothing}}}{\bar{Q}}$ |
|---|---|---|---|---|---|---|
| 5_gen_protoss | everything | $8.4 \pm 0.05$ | $1.5 \pm 0.05$ | $0.179 \pm 0.006$ | $1.18 \pm 0.04$ | $0.119 \pm 0.004$ |
| | ally_all_except_actions | | $0.69 \pm 0.02$ | $0.082 \pm 0.004$ | $0.52 \pm 0.01$ | $0.022 \pm 0.002$ |
| | ally_health_and_shield | | $0.62 \pm 0.02$ | $0.074 \pm 0.004$ | $0.45 \pm 0.02$ | $0.014 \pm 0.002$ |
| | ally_health | | $0.54 \pm 0.03$ | $0.064 \pm 0.005$ | $0.39 \pm 0.02$ | $0.004 \pm 0.001$ |
| | ally_distance | | $0.52 \pm 0.04$ | $0.062 \pm 0.006$ | $0.38 \pm 0.03$ | $0.0019 \pm 0.0005$ |
| | ally_shield | | $0.52 \pm 0.04$ | $0.062 \pm 0.006$ | $0.38 \pm 0.03$ | $0.0018 \pm 0.0008$ |
| | ally_all | | $0.85 \pm 0.009$ | $0.102 \pm 0.004$ | $0.656 \pm 0.01$ | $0.042 \pm 0.002$ |
| | enemy_health_and_shield | | $0.79 \pm 0.03$ | $0.094 \pm 0.004$ | $0.6 \pm 0.03$ | $0.034 \pm 0.003$ |
| | enemy_health | | $0.69 \pm 0.05$ | $0.082 \pm 0.007$ | $0.51 \pm 0.04$ | $0.022 \pm 0.003$ |
| | enemy_distance | | $0.56 \pm 0.03$ | $0.067 \pm 0.006$ | $0.41 \pm 0.03$ | $0.0074 \pm 0.0002$ |
| | enemy_shield | | $0.57 \pm 0.03$ | $0.068 \pm 0.005$ | $0.42 \pm 0.02$ | $0.008 \pm 0.002$ |
| | enemy_all | | $0.96 \pm 0.02$ | $0.116 \pm 0.004$ | $0.74 \pm 0.02$ | $0.055 \pm 0.002$ |
| | nothing | | $0.5 \pm 0.03$ | $0.06 \pm 0.006$ | $0.37 \pm 0.03$ | |
| 5_gen_terran | everything | $8.2 \pm 0.1$ | $2.0 \pm 0.1$ | $0.243 \pm 0.009$ | $1.6 \pm 0.1$ | $0.148 \pm 0.008$ |
| | ally_all_except_actions | | $1.0 \pm 0.05$ | $0.1221 \pm 0.001$ | $0.75 \pm 0.04$ | $0.028 \pm 0.002$ |
| | ally_health_and_shield | | $0.82 \pm 0.04$ | $0.1 \pm 0.001$ | $0.6 \pm 0.03$ | $0.0052 \pm 0.0009$ |
| | ally_health | | $0.83 \pm 0.04$ | $0.1007 \pm 0.0003$ | $0.6 \pm 0.04$ | $0.0061 \pm 0.0007$ |
| | ally_distance | | $0.78 \pm 0.04$ | $0.0945 \pm 9e-05$ | $0.56 \pm 0.03$ | $0.0002 \pm 0.0007$ |
| | ally_shield | | $0.75 \pm 0.02$ | $0.092 \pm 0.002$ | $0.55 \pm 0.02$ | $-0.003 \pm 0.001$ |
| | ally_all | | $1.18 \pm 0.06$ | $0.145 \pm 0.002$ | $0.92 \pm 0.06$ | $0.05 \pm 0.003$ |
| | enemy_health_and_shield | | $0.98 \pm 0.06$ | $0.12 \pm 0.003$ | $0.72 \pm 0.05$ | $0.025 \pm 0.003$ |
| | enemy_health | | $0.97 \pm 0.07$ | $0.118 \pm 0.003$ | $0.71 \pm 0.05$ | $0.023 \pm 0.003$ |
| | enemy_distance | | $0.82 \pm 0.04$ | $0.1002 \pm 0.0002$ | $0.59 \pm 0.03$ | $0.0055 \pm 0.0006$ |
| | enemy_shield | | $0.75 \pm 0.05$ | $0.091 \pm 0.005$ | $0.54 \pm 0.04$ | $-0.004 \pm 0.004$ |
| | enemy_all | | $1.2 \pm 0.1$ | $0.151 \pm 0.007$ | $0.95 \pm 0.09$ | $0.057 \pm 0.008$ |
| | nothing | | $0.78 \pm 0.03$ | $0.0945 \pm 0.0007$ | $0.56 \pm 0.03$ | |
| 5_gen_zerg | everything | $7.2 \pm 0.05$ | $2.41 \pm 0.05$ | $0.33 \pm 0.02$ | $1.86 \pm 0.04$ | $0.176 \pm 0.009$ |
| | ally_all_except_actions | | $1.36 \pm 0.02$ | $0.188 \pm 0.008$ | $0.99 \pm 0.02$ | $0.031 \pm 0.002$ |
| | ally_health_and_shield | | $1.15 \pm 0.03$ | $0.16 \pm 0.005$ | $0.83 \pm 0.03$ | $0.002 \pm 0.001$ |
| | ally_health | | $1.17 \pm 0.03$ | $0.161 \pm 0.005$ | $0.83 \pm 0.02$ | $0.0037 \pm 0.0008$ |
| | ally_distance | | $1.18 \pm 0.02$ | $0.164 \pm 0.007$ | $0.85 \pm 0.02$ | $0.006 \pm 0.001$ |
| | ally_shield | | $1.13 \pm 0.02$ | $0.157 \pm 0.006$ | $0.81 \pm 0.02$ | $-0.0005 \pm 0.0003$ |
| | ally_all | | $1.46 \pm 0.02$ | $0.202 \pm 0.01$ | $1.09 \pm 0.02$ | $0.044 \pm 0.002$ |
| | enemy_health_and_shield | | $1.31 \pm 0.03$ | $0.18 \pm 0.008$ | $0.95 \pm 0.03$ | $0.024 \pm 0.002$ |
| | enemy_health | | $1.3 \pm 0.02$ | $0.179 \pm 0.007$ | $0.94 \pm 0.02$ | $0.022 \pm 0.001$ |
| | enemy_distance | | $1.24 \pm 0.01$ | $0.173 \pm 0.008$ | $0.88 \pm 0.02$ | $0.014 \pm 0.002$ |
| | enemy_shield | | $1.14 \pm 0.02$ | $0.159 \pm 0.006$ | $0.82 \pm 0.02$ | $0.0005 \pm 0.0008$ |
| | enemy_all | | $1.67 \pm 0.01$ | $0.23 \pm 0.01$ | $1.24 \pm 0.02$ | $0.074 \pm 0.008$ |
| | nothing | | $1.14 \pm 0.02$ | $0.157 \pm 0.007$ | $0.81 \pm 0.02$ | |

of 20 episodes every 2k steps. The hyperparameters used for QMIX are given in Table 4, for MAPPO and IPPO in Table 5, and for QPLEX in Table 6.

## C.4 EPO Runs

This section provides the implementation details for the Extended Partial Observability challenge in SMACv2 and outlines the corresponding training procedure.

For the EPO baselines in Section 7.2, we used the same hyperparameters as Section 7.1. Implementations of MAPPO and QMIX were also consistent with those in Section 7.1. Each training run and corresponding evaluations were conducted across 3 random seeds. Experiments for this section were conducted on 80-core CPU machines with NVIDIA GeForce RTX690 2080 Ti or Tesla V100-SXM2-16GB GPUs. Both MAPPO and QMIX experiments took between 36-48 hours to complete, each running on 8 GPUs, for 3 $p$ values, across 3 seeds and 3 races.

# D Environment Additional Details

## D.1 SMACv2 Additional Details

In this section we describe the generation of teams in SMACv2. For each race, 3 different types of units are used. Each race has a special unit that should not be generated too often. For Protoss this is the colossus. This is a very powerful unit. If a team has many colossi, the battle will devolve into a war about who can use their colossi most effectively. Terran has the medivac unit. This is a healing unit that cannot attack the enemy and so is only spawned sparingly. Zerg has the baneling unit. This is a suicidal unit which deals area-of-effect damage to enemy units by rolling into them and exploding. In scenarios with many banelings, the agents learn to spread out and hide in the corners

Table 4: QMIX hyperparameters used for experiments. Parameters with (SMAC) or (SMACv2) after them denote that parameter setting was only used for SMAC or SMACv2 experiments respectively. These are the values in the corresponding configuration file in PyMarl[37, 13]. Mac is the code responsible for marshalling inputs to the neural networks, learner is the code used for learning and runner determines whether experience is collected in serial or parallel.

| Parameter Name | Value |
| --- | --- |
| Action Selector | epsilon greedy |
| $\epsilon$ Start | 1.0 |
| $\epsilon$ Finish | 0.05 |
| $\epsilon$ Anneal Time | 100000 |
| Runner | parallel |
| Batch Size Run | 4 |
| Buffer Size | 5000 |
| Batch Size | 128 |
| Optimizer | Adam |
| $t_{\max}$ | 10050000 |
| Target Update Interval | 200 |
| Mac | n_mac |
| Agent | n_rnn |
| Agent Output Type | q |
| Learner | nq_learner |
| Mixer | qmix |
| Mixing Embed Dimension | 32 |
| Hypernet Embed Dimension | 64 |
| Learning Rate | 0.001 |
| $\lambda$ (SMAC) | 0.6 |
| $\lambda$ (SMACv2) | 0.4 |

with the hope that the enemy banelings explode and the allies win by default. All of these special units are spawned with a probability of 10%. The other units used spawn with a probability of 45%. This is summarised in Table 9.

There are two changes to the observation space from SMAC. First, each agent observes their own field-of-view direction. Secondly, each agent observes their own position in the map as x- and y-coordinates. This is normalised by dividing by the map width and height respectively. The only change to the state from SMAC was to add the field-of-view direction of each agent to the state.

Additionally, we made one small change to the reward function in SMACv2. This was to fix a bug where the enemies healing can give allied units reward. There are more details available about this problem in the associated Github issue. SMACv2 also has an identical API to the original SMAC, allowing for very simple transition between the two frameworks.

The code for SMACv2 can be found in the Github repo, where there is a README detailing how to run the benchmark with random agents.

### D.2    EPO Additional Details

In EPO, the first ally agent to observe an enemy is guaranteed to observe the usual SMAC features associated with that enemy. Upon an enemy being observed for the first time, by any agent on the ally team, a random binary draw occurs for all other agents (i.e., those that had not yet observed this particular enemy). Random tie breaking ensures only one agent is guaranteed to see the enemy should two or more observe it for the first time on the same timestep. The draw is weighted by a tunable environmnet parameter $p$ corresponding to the probability of success. If the draw is successful for a particular agent, any future instances of the agent observing the enemy will occur as normal, without masking. If the draw is unsuccessful, that agent will not be able to observe the enemy on future timesteps, irrespective of whether or not it is within sight range. If the first agent to have observed the enemy dies, the next time the enemy falls within an ally agent's sight range that agent is guaranteed to see the enemy and the random draw occurs again for all other agents.

Table 5: MAPPO and IPPO hyperparameters used for the experiments on SMAC and SMACv2. Parameters with (SMAC) or (SMACv2) after them denote that parameter setting was only used for SMAC or SMACv2 experiments respectively.

| Hyperparameter | Value |
|---|---|
| Action Selector | multinomial |
| Mask Before Softmax | True |
| Runner | parallel |
| Buffer Size | 8 |
| Batch Size Run | 8 |
| Batch Size | 8 |
| Target Update Interval | 200 |
| Learning Rate Actor (SMAC) | 0.001 |
| Learning Rate Actor (SMACv2) | 0.0005 |
| Learning Rate Critic | 0.001 |
| $\tau$ | 0.995 |
| $\lambda$ | 0.99 |
| Agent Output Type | pi_logits |
| Learner | trust_region_learner |
| Critic (MAPPO) | centralV_rnn_critic |
| Critic (IPPO) | decentral_rnn_critic |
| Detach Every | 20 |
| Replace Every | None |
| Mini Epochs Actor | 10 |
| Mini Epochs Critic | 10 |
| Entropy Loss Coeff | 0.0 |
| Advantage Calc Method | GAE |
| Bootstrap Timeouts | False |
| Surrogate Clipped | True |
| Clip Range | 0.1 |
| Is Advantage Normalized | True |
| Observation Normalized | True |
| Use Popart | False |
| Normalize Value | False |
| Obs Agent Id | True |
| Obs Last Action | False |

# E   Analysis of Changes in SMACv2

To investigate the impact of the changes introduced in SMACv2, we perform three ablation studies. We focus only on the 5 unit scenarios, and only train MAPPO on the ablations because it is significantly faster to train than QMIX. We use the same hyperparameters and setup as in Section 7.1. We ablate different team compositions by using a single unit type for each of the non-special units in each race. We investigate random start positions by only using the *surround* or *reflect* scenarios and by removing the random start positions entirely. We also ablate the effect of using the true unit ranges. Additionally, we investigate partial observability by comparing feed-forward and recurrent networks.

The results of these experiments are shown in Figure 12. The position ablations show that stochastic starting positions and hence stochasticity itself contributes greatly to the challenge of SMACv2. Without random start positions, MAPPO achieves win rates of close to 100%. Both *surround* and *reflect* scenarios have a similar win rate, suggesting similar difficulty. There does not appear to be additional difficulty from combining the two. This is logical as the scenarios can be disambiguated using start positions at the beginning of the episode.

Our experiments of unit type ablations, shown in Figure 12b, indicate that unit variety significantly impacts the difficulty of the tasks. All three races show large differences between the easiest unit distribution and the baseline, suggesting that a diverse range of units contributes to SMACv2's difficulty. In both the Zerg and Protoss scenarios, the melee-only scenarios are easier than the ablation

Table 6: QPLEX hyperparameters used for experiments on SMACv2.

| Hyperparameter | Value |
| --- | --- |
| Action Selector | epsilon_greedy |
| Epsilon Start | 1.0 |
| Epsilon Finish | 0.05 |
| Epsilon Anneal Time | 50000 |
| Runner | parallel |
| Batch Size Run | 4 |
| Buffer Size | 5000 |
| Batch Size | 128 |
| Optimizer | adam |
| Target Update Interval | 200 |
| Agent Output Type | q |
| Learner | dmaq_qatten_learner |
| Double Q | True |
| Mixer | dmaq_qatten |
| Mixing Embed Dim | 32 |
| Hypernet Embed | 64 |
| Adv Hypernet Layers | 1 |
| Adv Hypernet Embed | 64 |
| Td Lambda | 0.8 |
| Num Kernel | 4 |
| Is Minus One | True |
| Is Adv Attention | True |
| Is Stop Gradient | True |
| N Head | 4 |
| Attend Reg Coef | 0.001 |
| State Bias | True |
| Mask Dead | False |
| Weighted Head | False |
| Nonlinear | False |
| Burn In Period | 100 |
| Name | qplex_qatten_sc2 |

Table 7: QMIX hyperparameter tuning grid

| Hyperparameter | Values Tried |
| --- | --- |
| $\epsilon$ annealing time | $[100 \times 10^3, 500 \times 10^3]$ |
| $\lambda$ | $[0.6, 0.8, 0.4]$ |

Table 8: MAPPO hyperparameter tuning grid

| Hyperparameter | Value |
| --- | --- |
| Actor Learning Rate | $[0.0007, 0.0004, 0.0001]$ |
| Clip Range | $[0.15, 0.05, 0.1]$ |

Table 9: Unit types used per-race in the SMACv2 scenarios.

| Race | Unit Types | Probability of Generation |
|------|-----------|---------------------------|
| Terran | Marine | 0.45 |
| | Marauder | 0.45 |
| | Medivac | 0.1 |
| Zerg | Zergling | 0.45 |
| | Baneling | 0.1 |
| | Hydralisk | 0.45 |
| Protoss | Stalker | 0.45 |
| | Zealot | 0.45 |
| | Colossus | 0.1 |

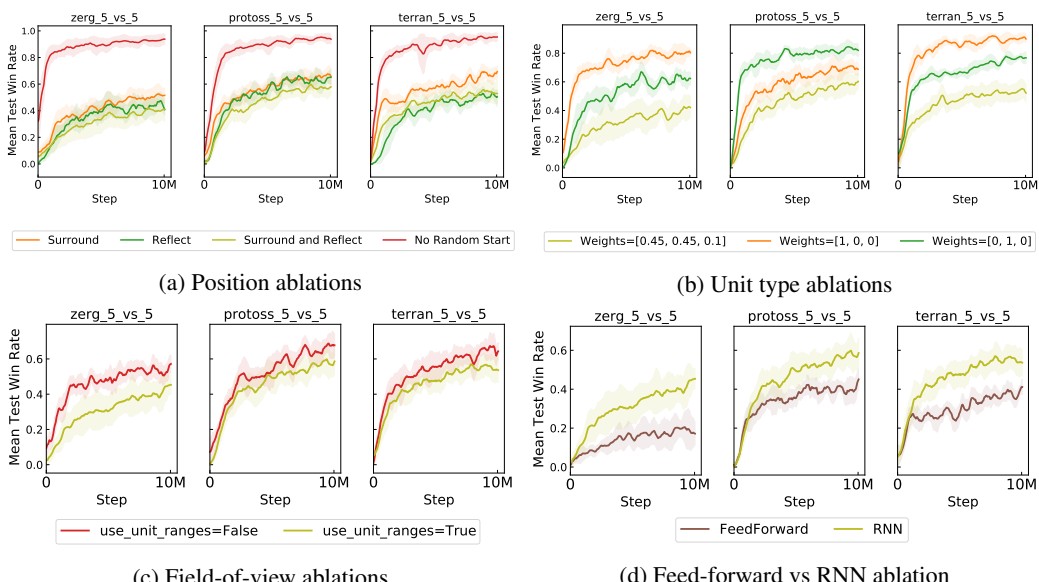

(a) Position ablations

(b) Unit type ablations

(c) Field-of-view ablations

(d) Feed-forward vs RNN ablation

Figure 12: Ablation of SMACv2 unit type, field-of-view and starting position features. Figure 12d shows the performance of a feed-forward policy compared to an RNN baseline. The unit weights are relative to a fixed order of units. For Zerg this is zergling, hydralisk, baneling, for Terran it is marine, marauder, medivac and for Protoss it is stalkers, zealots and colossi. Plots show mean and standard deviation across three seeds.

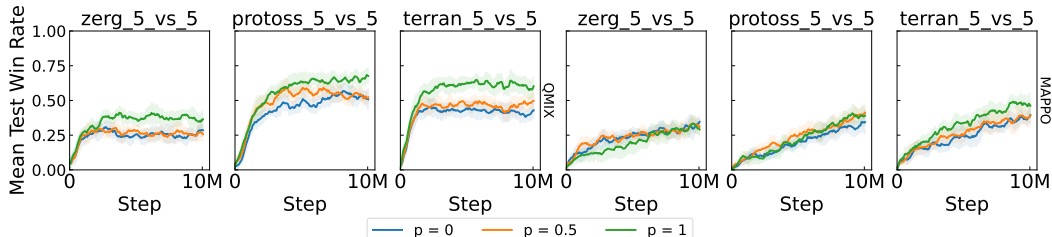

Figure 13: Comparison of the mean test win rate when $p = 0, 0.5, 1$ for QMIX (left) and MAPPO (right) on SMACv2. Here, the available action mask is still present.

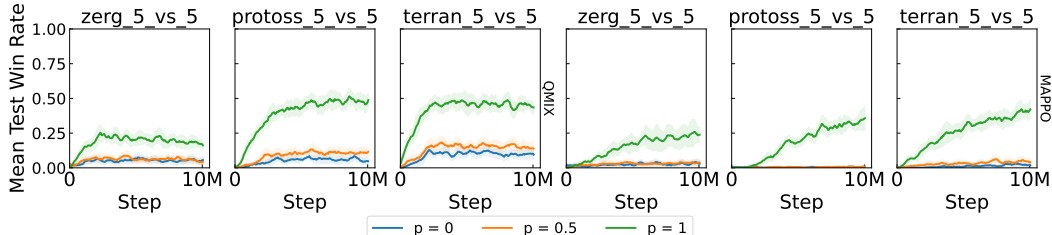

Figure 14: Comparison of mean test win rate when $p = 0, 0.5, 1$ with 5 agents against 5 enemy units, for QMIX (left) and MAPPO (right). Here, the available actions mask has been removed.

with ranged units. From observing episodes, the enemy AI tends to aggressively pursue allied units, which allows the allies to lure it out of position. This is easier to exploit with melee units than ranged ones. Terran scenarios have no melee units, but the marine scenarios are slightly easier. Overall these results show that generalising to different unit types is a significant part of the challenge in SMACv2.

Our field-of-view ablations, shown in Figure 12c, compare the fixed unit sight and attack ranges to the SMACv2 versions. There is an increase in difficulty from varying the unit ranges, but this effect is small. This suggests that the difficulty of SMACv2 is better explained by the diversity of start positions and unit types than the change to the true sight and attack ranges.

Finally, to evaluate the effect that the changes had on the partial observability of SMACv2, we compare a feedforward network with the performance of the RNN baseline, shown in Figure 12d. There is a significant decrease in performance from using the feedforward network. However, the feedforward network can still learn reasonable policies. This suggests that although being able to resolve partial observability is useful in SMACv2, the primary difficulty consists in generalising to a range of micro-management scenarios. This is in contrast to the extended partial observability challenge, where resolving partial observability through communication is the primary difficulty. The zerg scenario has a larger relative performance difference. Partial observability may be more of a problem here because the splash damage done to allies by banelings creates an incentive for allies to spread out more.

Overall, the results in Figures 12a and 12b show that current MARL methods struggle with the increased stochasticity due to map randomisation. To solve this problem, agents must be able to use their observations to make inferences about relevant state or joint observation features in the Dec-POMDP in more general settings. Current inference capabilities could be improved by research on more effective sequence models, such as specialised transformers [47, 16, 5]. The work in this area either applies transformer models to traditional RL methods [15], or focuses on paradigms similar to upside-down RL [39, 5]. While the latter work shows promising generalisation [34], it is mostly in the offline setting. Hu et al. [14] demonstrate a promising sequence model that can be applied to any MARL algorithm, but only demonstrate their work on marine units, and rely on the Dec-POMDP having a certain structure. Expanding such work to handle more diverse scenarios is an interesting avenue of future work.

The code for the ablations can be found here. This is distributed under an MIT license. The changes on this branch enable easy running of the ablations and add associated environment configurations for this purpose.

Table 10: SMACv2 Scenarios with their number of allies, number of enemies, and unit types.

| Scenario Name | Number of Allies | Number of Enemies | Unit Types |
|---|---|---|---|
| `protoss_5_vs_5` | 5 | 5 | Stalker, Zealot, Colossus |
| `protoss_10_vs_10` | 10 | 10 | Stalker, Zealot, Colossus |
| `protoss_20_vs_20` | 20 | 20 | Stalker, Zealot, Colossus |
| `protoss_10_vs_11` | 10 | 11 | Stalker, Zealot, Colossus |
| `protoss_20_vs_23` | 20 | 23 | Stalker, Zealot, Colossus |
| `terran_5_vs_5` | 5 | 5 | Marine, Marauder, Medivac |
| `terran_10_vs_10` | 10 | 10 | Marine, Marauder, Medivac |
| `terran_20_vs_20` | 20 | 20 | Marine, Marauder, Medivac |
| `terran_10_vs_11` | 10 | 11 | Marine, Marauder, Medivac |
| `terran_20_vs_23` | 20 | 23 | Marine, Marauder, Medivac |
| `zerg_5_vs_5` | 5 | 5 | Zergling, Hydralisk, Baneling |
| `zerg_10_vs_10` | 10 | 10 | Zergling, Hydralisk, Baneling |
| `zerg_20_vs_20` | 20 | 20 | Zergling, Hydralisk, Baneling |
| `zerg_10_vs_11` | 10 | 11 | Zergling, Hydralisk, Baneling |
| `zerg_20_vs_23` | 20 | 23 | Zergling, Hydralisk, Baneling |

## F   Limitations and Broader Impact

The main limitation of SMACv2 is that it is confined to scenarios within the game of StarCraft II. This is an environment which, while complex, cannot represent the dynamics of all multi-agent tasks. Evaluation of MARL algorithms therefore should not be limited to one benchmark, but should target variety with a range of tasks.

Whilst the scenarios in SMACv2 involve battles between two armies of units, only one side can be controlled by RL agents. It is technically possible to control two armies using two StarCraft II clients that communicate via LAN, which would allow to train two groups of decentralised agents against each other, e.g., via self-play. We leave the implementation of this functionality as future work.

SMACv2 aims to contribute to the development of MARL algorithms. As with any machine learning field, it is possible that improving the capabilities of these algorithms could lead to unethical uses. However, there are also many potential benefits to better cooperative AI, such as applications in automated driving among others. We believe that the potential benefits of developing more capable and cooperative AI outweigh the potential risks.

