# OpenReview forum: "SMACv2: An Improved Benchmark for Cooperative Multi-Agent Reinforcement Learning"
_NeurIPS.cc/2023/Track/Datasets_and_Benchmarks — NeurIPS 2023 Datasets and Benchmarks Poster_

### Official Review · Reviewer_sKYb · 2023-06-22
**Great work**

**Rating:** 9
**Confidence:** 3
**Correctness:** The made adjustments to SMAC seem rea…

**Strengths:**

- Uncovers a fundamental issue in a popular MARL benchmark
- Fixes it
- Clear experiments showing the flaw and that the fix seems to work

**Additional Feedback:**

-

**Clarity:**

Overall well written. The part about choosing $p$ might be a bit short/technical but is nonetheless comprehensible.

**Documentation:**

Well documented on GitHub.

**Ethics:**

No ethical concerns

**Limitations:**

-

**Opportunities For Improvement:**

I think it would be great if the paper included a more general discussion in the conclusion that RL/MARL research should not rely on a single benchmark environment due to, among others, the issues uncovered in the submission, i.e., one cannot prove the absence of bugs.

**Relation To Prior Work:**

Yes

**Summary And Contributions:**

The paper shows that the StarCraft Multi-Agent Challenge (SMAC), one of the dominant MARL benchmarks, is flawed due to a lack of stochasticity and partial observability. Specifically, the paper demonstrates that "open-loop" agents, i.e., that just remember action sequences, could match the performance of normal "closed-loop" agents in some scenarios of SMAC.
The paper then proposes SMACv2 by introducing randomized starting positions and randomness in agents being able to observe other agents (i.e., partial observability), and shows that open-loop policies fail to achieve a high reward on the new challenge.

---

> ### Author Response · Authors · 2023-08-09
> **Response to Reviewer sKYb**
>
> Thank you very much for your kind review. We have addressed your comments below. We hope that this clarifies any concerns that you had.
>
> >I think it would be great if the paper included a more general discussion in the conclusion that RL/MARL research should not rely on a single benchmark environment due to, among others, the issues uncovered in the submission, i.e., one cannot prove the absence of bugs.
>
> We strongly agree with this point! A meta-analysis of MARL evaluation found that a significant number of papers only used one environment to benchmark on (see [1] Figure 4(b)]). We have added a brief discussion highlighting this to the conclusion.
>
> >Overall well written. The part about choosing p might be a bit short/technical but is nonetheless comprehensible.
>
> Thank you for pointing this out. We have gone over the relevant section and made a few changes to improve readability.
>
> [1] Gorsane, Rihab, et al. "Towards a standardised performance evaluation protocol for cooperative marl." Advances in Neural Information Processing Systems 35 (2022): 5510-5521.

---

### Official Review · Reviewer_QzEt · 2023-07-19
**SMACv2 raises the bar for multi-agent reinforcement learning research.**

**Rating:** 8
**Confidence:** 4

**Strengths:**

This study is of excellent quality and would substantially benefit the RL research community. I very much look forward to seeing the next generation algorithms that can crack SMACv2!

The demonstration that open-loop algorithms, which were conditioned only on agent ids and timestep, can solve the SMAC was convincing and striking, highlighting the critical need for a new, challenging benchmark.

The introduction and successful validation of stochasticity in SMACv2, using the QMIX and MAPPO algorithms, confirmed that the new benchmark effectively addressed the weakness of SMAC.

The clever inclusion of the extended partial observability challenge in SMACv2, which promotes communication among agents, has opened up intriguing avenues for further research.


**Additional Feedback:**

References with incomplete information: [16], [22], [30], [34], [36]


**Clarity:**

This paper is written very clearly.


**Correctness:**

The authors convincingly demonstrated the weaknesses of SMAC and validated SMACv2 with well-designed experiments.

The Feature Inferrability and Relevance analyses seem promising, and providing more details would be helpful.


**Documentation:**

The SMACv2 benchmark, QMIX, and MAPPO are openly available with good documentation. However, I could not find the code for running the experiments in this study.


**Ethics:**

No.

**Limitations:**

I have specific suggestions.

The Feature Inferrability and Relevance sections (5.2 and 7.3) look interesting, but I had some trouble understanding. Specifically,
* In 7.3, important features (e.g., enemy health) were only briefly mentioned. If the detailed results for each mask can be presented in the style of Appendix Table 3 (SMACv2 feature quality experiment results, but only with everything and nothing masks), it would be very helpful.
* Different mask conditions in the main figure 3, and appendix figures 3-4 are not clearly distinguished, and I could not find what the y-axis “Episode Running” meant.
* “The Root-mean-squared error as a proportion of mean Q value” metric made more sense to me in comparing the effects of feature masking. If this metric was consistently used when comparing SMAC vs. SMACv2 (5.2 vs. 7.3), it would be helpful.
* The columns between the Appendix Tables 2 and 3 should be matched, by adding the last column of the Appendix Table 3 to Table 2.

When referring to the Appendix, clearly identify the relevant sections or tables to facilitate easy access and comprehension for readers.


**Opportunities For Improvement:**

I only have a minor comment. The authors should add potential negative societal impacts.

**Relation To Prior Work:**

The authors’ deep understanding of a SMAC led them to develop a much stronger benchmark, SMACv2.


**Summary And Contributions:**

The StarCraft Multi-Agent Challenge (SMAC) has been widely used as a benchmark for cooperative multi-agent reinforcement learning (MARL). However, recent works have achieved near-perfect performance on SMAC, raising concerns about its effectiveness as a challenging benchmark for MARL algorithms. In this study, the authors identified key weaknesses in SMAC, particularly its lack of stochasticity and meaningful partial observability. Strikingly, they demonstrated that an open-loop policy that only takes timestep, and no other observation, can perform well.

To address these limitations, the authors proposed SMACv2, a new benchmark that aims to introduce stochasticity and meaningful partial observability into the environment. SMACv2 challenges agents with procedurally generated team compositions and random start positions, necessitating the use of closed-loop policies for successful performance. Additionally, they introduced an extended partial observability (EPO) challenge, where agents must communicate enemy information to prioritize targets. The authors evaluated the QMIX and MAPPO algorithms on SMACv2 and confirmed that these models indeed struggled with the increased difficulty of the new benchmark.

Together, SMACv2 effectively raises the bar for MARL research, offering a more challenging environment for evaluating and advancing multi-agent learning algorithms.

---

> ### Author Response · Authors · 2023-08-09
> **Response to Reviewer QzEt**
>
> We would like to thank you very much for your review and your kind words about our paper. We are pleased that you found our paper raises the bar for MARL research.
>
> Below we have addressed your questions. We hope that this strengthens your support for the paper.
>
> >I only have a minor comment. The authors should add potential negative societal impacts.
>
> We have added some commentary about this to the paper addressing this. The changes are highlighted in red.
>
> >In 7.3, important features (e.g., enemy health) were only briefly mentioned. If the detailed results for each mask can be presented in the style of Appendix Table 3 (SMACv2 feature quality experiment results, but only with everything and nothing masks), it would be very helpful.
>
> Thank you for your comment. We have added these results to Tables 2 and 3 in the appendix.
>
> >Different mask conditions in the main figure 3, and appendix figures 3-4 are not clearly distinguished, and I could not find what the y-axis “Episode Running” meant. “The Root-mean-squared error as a proportion of mean Q value” metric made more sense to me in comparing the effects of feature masking. If this metric was consistently used when comparing SMAC vs. SMACv2 (5.2 vs. 7.3), it would be helpful.
>
> Thank you for your comment. The "episodes running" just refers to the proportion of episodes that have not yet terminated by that time-step in the episode. We have added a description of this to the caption of Figure 3 along with a clarification that the x-axis
> is episode steps.
>
> What is masked by each different mask is explained in Table 1 in the appendix. We have added a reference to this table in the main paper.
>
> Although the root-mean-squared error as a proportion of the mean Q-value is a useful metric, directly comparing it across
> environments is slightly flawed. This is because it ignores the aleatoric uncertainty in training neural networks. This is the
> purpose behind the *nothing* mask and by comparing SMAC and SMACv2 by looking at the difference between the *everything* and
> *nothing* masks.
>
> > The columns between the Appendix Tables 2 and 3 should be matched, by adding the last column of the Appendix Table 3 to Table 2.
>
> Thank you for pointing this out. We have updated Table 2 in the Appendix to reflect this change.
>
> > When referring to the Appendix, clearly identify the relevant sections or tables to facilitate easy access and comprehension for readers.
>
> Thank you for raising this point. We have updated the paper to make references to the Appendix specific.
>
> >The SMACv2 benchmark, QMIX, and MAPPO are openly available with good documentation. However, I could not find the code for running the experiments in this study.
>
> We have added RUNNING_EXPERIMENTS.md to the SMACv2 repo to point out what scripts to run to recreate the major results in the paper. This should help people to recreate the work better.
>
> > References with incomplete information: [16], [22], [30], [34], [36]
>
> Thank you very much for pointing this out. We have corrected these references.

---

### Official Review · Reviewer_9mMA · 2023-07-24
**well-written paper; minor questions**

**Rating:** 8
**Confidence:** 3

**Additional Feedback:**

Minor typos and language related changes: Line 146, full stop after "in the appendix"

**Clarity:**

For the experiments in Section 5, did the authors use the shaped reward or the sparse reward in SMAC \[1\] ? Could the authors clarify if the effectiveness of open-loop policies hold in the shaped reward case (if sparse reward was used for Section 5 experiments) ? In my understanding, Q function should depend on the reward function choice (due to its form), and in the shaped reward case, this means that Q value should heavily depend on the state information, right?

**Opportunities For Improvement:**

The authors discuss how SMAC \[1\] has been used in a lot of prominent MARL papers (line 36). However, for this paper, they only evaluate MAPPO and QMIX. It would have been good to know which other MARL approaches (listed on line 36) perform well with SMACv2, and are thus more reliable as baselines for other MARL papers.

References:

1. The StarCraft Multi-Agent Challenge, Samvelyan et al. (2019)

Update (19 Aug 2023): Increased score from 7->8.

**Summary And Contributions:**

In this work, the authors illustrate certain problems with the StarCraft testbed (SMAC) used frequently in co-operative MARL literature. These problems arise from the empirical observation that in certain scenarios of SMAC, an open-loop policy (which doesn't use any state information) is able to perform just as well as a closed-loop policy, demonstrated through experiments that mask some or all of the state features. These experiments make sense and are well motivated. To address these issues, the authors propose SMACv2, by modifying SMAC to introduce more random initial configurations, and by restricting the observations so that the agents have to collaborate better in order to find a good policy. I think that these changes improve the SMAC testbed and set a higher standard for MARL algorithms, necessitating learning a more general policy to actually succeed in these tasks. The authors further show that open-loop policies do not work with the proposed SMACv2 testbed.

---

> ### Author Response · Authors · 2023-08-09
> **Response to Reviewer 9mMA**
>
> Thank you very much for your review! We are pleased that you found the experiments demonstrating the lack of  stochasticity in SMAC compelling and that you thought we have set a higher standard for MARL algorithms.
>
> We address your concerns below and hope this will strengthen your support for the paper.
>
> >The authors discuss how SMAC [1] has been used in a lot
> >of prominent MARL papers (line 36). However, for this
> >paper, they only evaluate MAPPO and QMIX. It would have
> >been good to know which other MARL approaches (listed on
> >line 36) perform well with SMACv2, and are thus more
> >reliable as baselines for other MARL papers.
>
> Thank you for this comment. We are running IPPO [1] and QPLEX [2] as additional baselines and will update the paper with the results once we have them.
>
> >For the experiments in Section 5, did the authors use the shaped reward or the sparse reward in SMAC [1] ?
>
> The experiments throughout the paper use SMAC's shaped reward.
>
> > In my understanding, Q function should depend on the reward function choice (due to its form), and in the shaped reward case, this means that Q value should heavily depend on the state information, right?
>
> In QMIX, the joint Q-function, $\mathbf{Q}$, is a monotonic mixture of each per-agent utility function $Q_i$. However, to allow for decentralised execution, each agent's utility function can only condition on its observation $o_i$ at test time.
>
> QMIX requires $\mathbf{Q}$ to condition on the state *and* regress to the true joint Q-function. To achieve this, QMIX uses a hypernetwork which takes the state $s$ as input and outputs the weights for the mixing network, ensuring that at test time, the entire mixing network is not needed.
>
> In the context of Section 5, this means that for the open-loop training, **we only need to mask all the observations and not the state**. This is because at test time, any network that conditions on the state will be thrown away. In the case of MAPPO this is the critic, and in the case of QMIX it is this mixing hypernetwork.
>
> In Section 5.3 however, we are evaluating the regression performance of the **joint Q-function** $\mathbf{Q}$, and therefore we mask the relevant featurs in the **state and the observation** to avoid information leakage. We have added clarifications about these points to the main text.
>
> > Minor typos and language related changes: Line 146, full stop after “in the appendix”
>
> Thank you very much for noticing this. We have fixed the error.
>
>
> Thank you again for your review. We hope that this has addressed your concerns and strengthens your support for the paper.
>
> [1] de Witt, Christian Schroeder, et al. "Is independent learning all you need in the starcraft multi-agent challenge?." arXiv preprint arXiv:2011.09533 (2020).
>
> [2] Wang, Jianhao, et al. "QPLEX: Duplex Dueling Multi-Agent Q-Learning." International Conference on Learning Representations. 2020.

---

> > ### Comment · Reviewer_9mMA · 2023-08-19
> > **Rebuttal response**
> >
> > I am satisfied with the authors' response. I will update my score.

---

> > > ### Author Response · Authors · 2023-08-29
> > >
> > > Thank you very much for taking the time to respond to our rebuttal and for updating your score!

---

### Author Response · Authors · 2023-08-29

We have updated the paper with results from IPPO and QPLEX baselines. Unfortunately some of the experiments crashed and therefore we have not been able to attain full results for all of the maps. The missing results are noted in the figure caption. We will of course ensure that these missed runs are completed for the camera-ready submission.

---

### Decision · Program_Chairs · 2023-09-22

**Decision:**

Accept (Poster)

**Comment:**

This work points out certain problems with the StarCraft testbed (SMAC) used frequently in co-operative MARL literature and provides a new benchmark SMACv2, where scenarios are procedurally generated and require agents to generalize to previously unseen settings.
The authors further show that open-loop policies do not work well on the proposed SMACv2 testbed which poses new challenges. This work is solid and the authors will add promised new results to the revision to make it more comprehensive.